

# Assessing the different components of the water balance of Lake Titicaca

Nilo Lima-Quispe[1], Denis Ruelland[2], Antoine Rabatel[1], Waldo Lavado-Casimiro[3], Thomas Condom[1]

[1]Univ. Grenoble Alpes, IRD, CNRS, INRAe, Grenoble-INP, Institut Geosciences Environnement (IGE, UMR 5001), 460 rue
de la Piscine, 38058 Grenoble cedex 9, FRANCE
[2]CNRS, HydroSciences Laboratory, University of Montpellier, 15 avenue Flahault, 34093 Montpellier Cedex 5, FRANCE
[3]Servicio Nacional de Meteorología e Hidrología (SENAMHI), 15072 Lima, PERU

*Correspondence to*: Nilo Lima-Quispe (nilo-roberto.lima-quispe@univ-grenoble-alpes.fr)

**Abstract.** This study estimates the water balance of a poorly-gauged large lake using an integrated modeling framework that accounts for natural hydrologic processes and net irrigation consumption. The modeling framework was tested on Lake Titicaca, located in the Altiplano of the Central Andes of South America. We used a conceptual approach based on the Water Evaluation and Planning System (WEAP) platform at a daily time step for the period 1982–2016, considering the following terms of the water balance: upstream inflows, direct precipitation and evaporation over the lake, and downstream outflows. To estimate upstream inflows, we evaluated the impact of snow and ice processes and net irrigation withdrawals on predicted streamflow and lake water levels. We also evaluated the role of heat storage change in evaporation from the lake. The results showed that the proposed modeling framework makes it possible to simulate lake water levels ranging from 3,808 to 3,812 m a.s.l. with good accuracy ($RMSE = 0.32$ m d$^{-1}$) under a wide range of long-term hydroclimatic conditions. The estimated water balance of Lake Titicaca shows that upstream inflows account for 56% (958 mm yr$^{-1}$) and direct precipitation over the lake for 44% (744 mm yr$^{-1}$) of the total inflows, while 93% (1,616 mm yr$^{-1}$) of total outflows are due to evaporation and the remaining 7% (121 mm yr$^{-1}$) to downstream outflows. The water balance closure has an error of -15 mm yr$^{-1}$. At the scale of the Lake Titicaca catchment, snow and ice processes, and net irrigation withdrawals had minimal impact on predicted upstream inflow. Thus, Lake Titicaca is primarily driven by variations in precipitation and high evaporation rates. The proposed modeling framework could be replicated in other poorly-gauged large lakes, as we demonstrate that a simple representation of natural hydrologic processes and irrigation enables accurate simulation of water levels.

## 1 Introduction

### 1.1 Lake water level fluctuations and water balance

Lakes are water reservoirs of vital importance for the development of regions as they provide many ecosystem services, including fisheries, water supply, tourism and energy generation (Sterner et al., 2020). However, these services can be impacted by fluctuating lake water levels. For instance, Yao et al. (2023) showed that, in the period from 1992 to 2020, there was a significant decrease in water levels in 43% of natural lakes (457), an increase in 22% (234), and non-significant trends were



observed in 35% (360). Knowing the main drivers of fluctuations in water levels is crucial for better lake management. This knowledge can be obtained by assessing the water balance, by quantifying water inflows, outflows, and storage change over a given period of time. Ideally the water balance is only based on measured data. However, since all flows in the hydrological cycle are far from being measured, estimating the components of the lake water balance, including direct precipitation and

evaporation, upstream inflows, downstream outflows, net groundwater exchanges and storage change, remains a complex challenge.

## 1.2 Methods to estimate the individual components of a lake water balance

Direct precipitation over a lake can be estimated using interpolation methods based on data recorded by ground stations (Chebud and Melesse, 2009), remotely-sensed observations (Vanderkelen et al., 2018) and meteorological reanalysis (Minallah

and Steiner, 2021). Interpolation methods are generally based on stations most of which are located in the surrounding areas (Kizza et al., 2012). This can lead to significant inaccuracies because large lakes (>500 km$^2$) have a major influence on regional climate (Scott and Huff, 1996; Su et al., 2020). For example, it has been observed that direct precipitation over many lakes is more intense than in their surrounding area (Anyah et al., 2006; Kizza et al., 2012; Nicholson, 2023; Thiery et al., 2015). According to Scott and Huff (1996) this is due to the differences in heat capacities between the lake surfaces and the

surrounding area, and the large amount of moisture that lakes provide to the lower atmosphere, which can lead to increased cloudiness and precipitation over lakes (Scott and Huff, 1996). Regarding the remotely-sensed datasets, Vanderkelen et al. (2018) obtained, for example, satisfactory precipitation estimates using Precipitation Estimation from Remotely Sensed Information using Artificial Neural Networks - Climate Data Record (PERSIANN-CDR) for Lake Victoria in Africa. On the other hand, in South America, Satgé et al. (2019) showed that 12 satellite-based products underestimated precipitation in Lake

Titicaca. Similarly, Hong et al. (2022) noted the shortcomings of satellite products in reliably representing precipitation over the Great Lakes in North America due to the lack of precipitation gauging over the lakes to calibrate the satellite images. Regarding meteorological reanalysis, Minallah and Steiner (2021) showed that the ECMWF Reanalysis version 5 (ERA5) (Hersbach et al., 2020) captured synoptic-scale precipitation patterns in three regions with the largest lakes in the world (Laurentian Great Lakes, Lake Baikal, and African Great Lakes) and that areas with intense precipitation coincided with the

presence of those lakes. However, other studies based on ERA5 (e.g. Xu et al., 2022) reported overestimation of precipitation over the Great Lakes of North America.

Direct evaporation from the lakes depends not only on meteorological conditions, but also on the size of the lake, water depth and water clarity, which all influence the energy balance due to changes in water temperature and vertical mixing (Lenters et al., 2005). Thus, estimating lake evaporation based on meteorological data alone can lead to inaccuracies (Bai and Wang,

2023). The best known methods include eddy covariance (Blanken et al., 2000; Liu, 2023; Shao et al., 2020), energy balance (Croley II, 1989; Gianniou and Antonopoulos, 2007; Lenters et al., 2005; Sturrock et al., 1992; Wang et al., 2014; Yu et al., 2011), Penman (Penman, 1948, 1956) and mass transfer (Dalton, 1802). Eddy covariance enables direct measurements of evaporation (Blanken et al., 2000), but few lakes have this type of instrumentation (Vesala et al., 2012). The energy balance



method is considered to be one of the most appropriate and accurate for estimating evaporation (Lenters et al., 2005), but

requires large quantities of data, meaning it is generally difficult to implement. The original Penman formulation, which does not include changes in heat storage, has been used to estimate lake evaporation (e.g. Kebede et al., 2006; Lima-Quispe et al., 2021). However, Blanken et al. (2011) observed a 5-month delay between peak net radiation and evaporation due to heat storage in Lake Superior in North America. One of the limitations of estimating the change in heat storage is the lack of water temperature data at different depths, and so models are used to simulate the thermal stratification dynamics of the water (e.g.

Antonopoulos and Gianniou, 2003).

For upstream inflow, ideally, measured streamflow data will be available. However, there are always areas in the catchments that contribute to the lakes that are ungauged (Wale et al., 2009). In addition, historical records of measured streamflow may have many missing or inconsistent data (Tencaliec et al., 2015). An alternative to generating continuous time series in gauged and ungauged catchments is using hydrological models driven by climate data and physiographic characteristics, calibrated

and evaluated against any available measured streamflow (e.g. Rientjes et al., 2011; Zhang and Post, 2018). For ungauged catchments, regionalization methods are applied based on the parameters obtained in the gauged catchments (e.g. Guo et al., 2021). Rientjes et al. (2011) addressed the parameter regionalization of the Lake Tana catchment using the HBV-96 model, where 62% of the area is ungauged. Similar studies (e.g. Wale et al., 2009) have been conducted to provide a methodological basis for estimating upstream inflow in both gauged and ungauged catchments. However, basic lumped rainfall-runoff

simulations may not be sufficient because the catchments that contribute to the lake involve very large areas with varied topography, and many natural hydrological processes are involved, as well as water uses. This means that modeling that integrates both natural hydrological processes and water uses may be necessary in hydrosystems undergoing significant anthropogenic pressure (e.g. Ashraf Vaghefi et al., 2015; Fabre et al., 2015; Hublart et al., 2016). In high mountain catchments, snow and ice processes can play a very important role in the hydrological response of catchments. For example, Li et al. (2020)

showed that although the glacierized area account for only 21% of the Kumalak catchment in Asia, snowmelt and ice melt account for 43% of total catchment runoff. Moreover, accelerated glacier retreat linked with global warming can are increasing catchment runoff, and several studies (e.g. Song et al., 2014; Sun et al., 2018) have reported the expansion and emergence of new lakes due to accelerated glacier retreat in the Tibetan Plateau. Estimating melt in glacierized upstream catchments may be key to a better understanding of the drivers of lake fluctuations. However, estimating accumulation and ablation processes is

complex because it is difficult to obtain accurate forcing data (e.g. precipitation and temperature) in high elevation areas where the measurement network is very sparse (Ruelland, 2020), as well as control data (e.g. upstream-area streamflow and glacier mass balance). In this context, temperature-index approaches (Hock, 2003) are more suitable than energy balance approaches and can produce simulations with acceptable accuracy using a reduced number of parameters and forcing data (e.g. Ruelland, 2023). In terms of water use, according to Wu et al. (2022), 60% of freshwater withdrawals worldwide are made for agricultural

irrigation. Accounting for net irrigation use in irrigated catchments may thus be critical. Catchment scale irrigation has been addressed using approaches based on soil water deficit (Kannan et al., 2011; Shadkam et al., 2016) and those that additionally consider irrigation scheduling (Githui et al., 2016; McInerney et al., 2018). Regardless of the approach chosen, one of the





limitations is the lack of measured irrigation data (McInerney et al., 2018), which hinders the evaluation of irrigation simulations. This evaluation can consequently only be undertaken indirectly by attempting to more realistically reproduce
observed outlet discharges by accounting for net consumption for water uses within the catchments (e.g. Fabre et al., 2015; Hublart et al., 2016). Nevertheless, in some regions, even spatial and temporal information on irrigable area and irrigated area is barely available (Lima-Quispe et al., 2021), making estimating the impact of water use very uncertain. Approaches based on soil water deficit are relatively easy to implement because they require few data (e.g., reference evapotranspiration, irrigable area, and crop phenology). However, some studies (e.g. Githui et al., 2016) argue that this type of approach simulates
unrealistic water withdrawals and that consistent simulations can only be obtained by including irrigation scheduling. Nevertheless, in a French catchment, Soutif-Bellenger et al. (2023) demonstrated that approaches based on soil water deficit can produce simulations with acceptable accuracy. Similarly, Hublart et al. (2016) obtained reliable simulations in an Andean catchment in Chile.

Net groundwater exchanges are another component of lake water balance that are difficult to quantify (Rosenberry et al., 2015).
They are neglected in some studies under the assumption that these fluxes are very small (Duan et al., 2018; Lima-Quispe et al., 2021). The interaction between groundwater and lakes has been addressed using conceptual (Parizi et al., 2022) and physically-based models (Vaquero et al., 2021; Xu et al., 2021), chemical and isotopic balances (Bouchez et al., 2016), and the water balance (Chavoshi and Danesh-Yazdi, 2022). The water balance is a fairly easily option, as the net flux is the results of the other components. However, it is crucial to dispose of accurate measurements or estimates of the other water balance
terms to avoid propagating uncertainty (Chavoshi and Danesh-Yazdi, 2022). Hydrochemical or isotopic analyses are considered accurate (Bouchez et al., 2016), but can be very costly for a large lake. Modeling approaches are often limited by data availability (Barthel and Banzhaf, 2016), especially when models are used that rely on physical laws and dynamically simulate the interaction between surface water and groundwater (Xu et al., 2021). On the other hand, downstream outflow is a component of the water balance to take into consideration in exorheic lakes that can be estimated in several ways including
direct measurements (Chebud and Melesse, 2009), the use of a rating curve relating lake level to outflow (Sene and Plinston, 1994), and as a residual of other water balance terms (Duan et al., 2018).

**1.3 Integrated water balance in large lakes**

In addition to the challenges involved in estimating the individual components of the water balance of large lakes, considering these processes in an integrated modeling framework is even more challenging and complex. The complexity increases when
the aim is to include water uses in upstream catchments. Several studies (e.g. Rientjes et al., 2011; Vanderkelen et al., 2018; Wale et al., 2009) have estimated the water balance under the assumption that net water withdrawals are negligible. However, this may not be valid in the future due to changing climate conditions and increased competition over water use, which could result in a decrease in upstream inflow (Wurtsbaugh et al., 2017). For example, in Lake Urmia, Schulz et al. (2020) showed that net withdrawals for irrigation exacerbate the decline in storage caused by climate change. A credible water balance
considering both natural flows and water uses is needed to identify the drivers of fluctuations in water level (Wurtsbaugh et



al., 2017). Few studies have attempted to address an integrated water balance in lake hydrosystems (e.g. Niswonger et al., 2014; Hosseini-Moghari et al., 2020). Niswonger et al. (2014) evaluated several management scenarios for the restoration of Walker Lake in the USA as part of a collaborative process with stakeholders. These authors implemented a modified version of the GSFLOW hydrologic model to dynamically simulate surface-groundwater interactions, reservoir management, water
withdrawals, irrigation, and total concentrations of dissolved solids. Although Walker Lake is a small lake (<500 km$^2$), this approach could be replicated for large lakes, but heterogeneity in data availability in space and over time is often a limitation. Hosseini-Moghari et al. (2020) ran the global water balance model, WaterGAP, with locally measured data to quantify the contributions of climate variability and human activities to the drop in the water level of Lake Urmia in Iran. However, their study mainly focused on estimating upstream inflow, with less emphasis on direct precipitation over the lake and evaporation,
which both play very important roles in the water balance (Gronewold et al., 2016). In addition, Hosseini-Moghari et al. (2020) used a simulation period of 11 years, which may be too short to understand the rise and fall dynamics of large lakes in the long term (Gronewold et al., 2013). To better manage large lakes, it is crucial to understand how long-term climate variability and anthropogenic pressures affect water level fluctuations. This could help identify the main drivers of lake fluctuations over time. However, in poorly-gauged regions, this task is complicated by heterogeneous data availability (Duan et al., 2018). The task
is even more challenging in the case of transboundary lake hydrosystems, where hydrometeorological monitoring is not always coordinated (Gronewold et al., 2018). Therefore, seeking a good trade-off between data availability and complexity in the representation of processes, without compromising the accuracy of simulations, remains an under-researched area in the context of long-term integrated water balance in large lakes.

## 1.4 Scope and objectives

The objective of the present study is to simulate the components of the water balance of a poorly-gauged large lake in an integrated modeling framework capable of capturing variations in water level under a wide range of long-term hydroclimatic conditions. The modeling framework was tested using Lake Titicaca, which is located in the Altiplano of the Central Andes of South America. This study area presents interesting challenges, as it is a poorly-gauged transboundary hydrosystem (Peru and Bolivia), with high mountain hydrological processes under a semi-arid tropical climate, with extreme climate variability,
and where irrigation is the main water use. To our knowledge, only Lima-Quispe et al. (2021) estimated the water balance of Lake Titicaca. However, their study has limitations, such as neglecting snow and ice processes, and the fact their estimation of lake evaporation using Penman's method did not consider heat storage change. In addition, evaporation was estimated using monthly averaged climate data (humidity, wind speed, and solar radiation), meaning it does not capture the physical processes and is not suitable for non-stationary conditions in the face of a changing climate. We provide new insights into the following
aspects of the Lake Titicaca water budget: (i) the impact of snow and ice processes, and of net irrigation withdrawals on the prediction of daily streamflow and lake water levels; (ii) the importance of including heat storage change for a more accurate estimation of evaporation from the lake; and (iii) an integrated water balance approach capable of accurately predicting the amplitude and frequency of long-term water level fluctuations at a daily time step.



## 2 Material

### 2.1 The Lake Titicaca hydrosystem

Lake Titicaca is located at 3,812 m a.s.l. in the Altiplano of the central Andes of South America. It is approximately 8,340 km$^2$ in size and crosses the borders of Peru and Bolivia. The elevation of the catchments that contribute to the lake ranges between 3,812 and 6,300 m a.s.l. (average 4,200 m a.s.l.) and cover an area of approximately 48,780 km$^2$. The lake has an average volume of 958 km$^3$ and a maximum depth of 277 m according to the bathymetry carried out between 2016 and 2019 (Autoridad Binacional del Lago Titicaca, 2021). Lake Titicaca belongs to a wide endoreic catchment, and is connected by the Desaguadero river to Lake Poopo in Bolivia (see Fig. 1). Lake Titicaca is of regional hydrological importance and the outflows of Lake Titicaca represent up to 79% of the inflows of Lake Poopo (Lima-Quispe et al., 2021). As a result, a significant reduction in the water level of Lake Titicaca can have a significant impact on inflows into Lake Poopo, as was the case in the 1990s (Lima-Quispe et al., 2021; Pillco Zola and Bengtsson, 2006). This has serious consequences for downstream water supplies for irrigation and fishing, and has led to high levels of emigration and impoverishment (Perreault, 2020).

The main challenges for the management of the lake are extreme hydrological events (droughts and floods), proper operation of lake releases, and pollution of the water bodies (Revollo, 2001; Rieckermann et al., 2006). Measurements in Puno (Peru) show that the water level has fluctuated over a range of about 6 m in the last century. The lowest levels were measured in 1943 and 1944. The highest levels were measured between 1984 and 1986 due to heavy precipitation that caused major flooding in the surrounding plain and downstream of the lake. Flooding caused damage estimated at $125 million (Revollo, 2001). As a transboundary lake, one of the main challenges is the use of water resources by Peru and Bolivia (Revollo, 2001). On the Peruvian side, the priority is maintaining an acceptable water level for the development of tourism, fishing and wetlands. On the Bolivian side, in addition to the above-mentioned priorities, it is essential to guarantee downstream releases for irrigation and the maintenance of Lake Poopó. Reconciling the priorities of each country is certainly an issue that needs to be considered.

In addition to the challenges associated with the quantity of water, the importance of water quality should not be overlooked. Large cities are located in the upstream catchment of the lake. On the Bolivian side is the city of El Alto with more than 1 million inhabitants. On the Peruvian side, the cities of Puno (125,018) and Juliaca (217,743) had about 342,761 inhabitants in 2017. These cities lack efficient sewage treatment, resulting in pollution of the rivers that eventually flow into Lake Titicaca (Archundia et al., 2016; Maldonado et al., 2023).

To address these challenges, a management plan for water resources was formulated in the early 1990s, not only for Lake Titicaca, but for the entire water system of the Altiplano (Revollo, 2001). This plan aimed to improve drought and flood management and to design and implement projects to satisfy water requirements. One of the main agreements was the construction of a dam to regulate releases from the lake and the establishment of operating rules. The dam was completed in 2001, but the lake releases are still almost the same as under natural conditions because the operating rules have not yet been applied. The Lake Titicaca Authority (Spanish acronym ALT) was also created as an autonomous binational entity with the mission of implementing the master plan.



**Figure 1: Main geographical features of the Lake Titicaca hydrosystem and location of the main streamflow gauges. The reference year for the limits of the glacier is 2000 (RGI Consortium, 2017). The reference year for croplands is 2010 (Ministerio de Desarrollo Rural y Tierras, 2010; Ministerio del Ambiente, 2015).**




## 2.2 Climate data

Daily precipitation and air temperature (see Fig. 2) were obtained from the data generated in Bolivia (Ministerio de Medio Ambiente y Agua, 2018) with the gridded meteorological ensemble tool (GMET) (Clark and Slater, 2006; Newman et al., 2015). GMET has a spatial resolution of 0.05° for the period 1980–2016. It is based on a probabilistic method using ground
station data (for more details see Clark and Slater (2006), and Newman et al. (2015)). GMET is run sequentially in two stages, where the first stage performs a locally-weighted spatial regression for each day, including considering variations in elevation of the terrain. In the second stage, ensemble members are created from the spatial regression to sample the regression uncertainty. For Bolivia, 30 ensemble members were generated for the period 1980–2016. The mean ensemble was used in the present study. GMET–Bolivia (hereafter GMET) considered ground stations on the Peruvian side and the spatial domain
consequently also includes the Peruvian catchments. In some catchments in the study area, Satgé et al. (2019) evaluated 12 satellite-based products and found that MSWEP and CHIRPS products were the most promising at the 10-day time step. As a result, for daily precipitation, four datasets including GMET, MSWEP, CHIRPS, and basic interpolation of ground station data with inverse distance weighting (IDW) (Ruelland, 2020) were initially tested according to daily hydrological sensitivity analyses. The results showed that GMET led to more accurate simulations in most catchments and Lake Titicaca (not shown
here for the sake of brevity).

Figure 2 shows the spatial pattern of precipitation (Fig. 2a) and air temperature (Fig. 2b) based on GMET for the period 1980–2016. Annual precipitation varies between 440 and 1,100 mm (mean 725 mm). The wettest areas are concentrated in the northwest and over Lake Titicaca. The driest areas are to the south. The spatial distribution does not show generalized dependence on elevation, but there are small areas on the eastern and western margin where precipitation increases with
elevation. The annual mean air temperature varies between -2°C and 11°C (average 6°C). The coldest areas are on the western and eastern areas, coinciding with the highest mountains. The warmest areas are located over Lake Titicaca. The spatial distribution of air temperature depends on elevation.

Daily data on relative humidity, wind speed, and solar radiation are very scarce in space and over time. Many values were missing in the time series from the weather stations, thus calling their quality and representativeness into question. For this
reason, we used reanalysis data from ERA5-Land (Muñoz-Sabater et al., 2021) which is available from 1950 to the present at a spatial resolution of 0.1° and a hourly time resolution. Data were aggregated at a daily time step for the present study. The quality of the ERA5-Land dataset was evaluated using data recorded at a station located at El Alto airport as it was the only one with reliable humidity and wind speed data and relatively long time series. For relative humidity, the performance was acceptable, as ERA5-Land was able to adequately reproduce both the annual magnitudes and the seasonal cycle. For wind
speed, ERA5-Land satisfactorily represented the seasonal cycle but with a significant systematic bias . The bias evaluated at the El Alto airport was used to correct the wind speed in the spatial domain of the study.



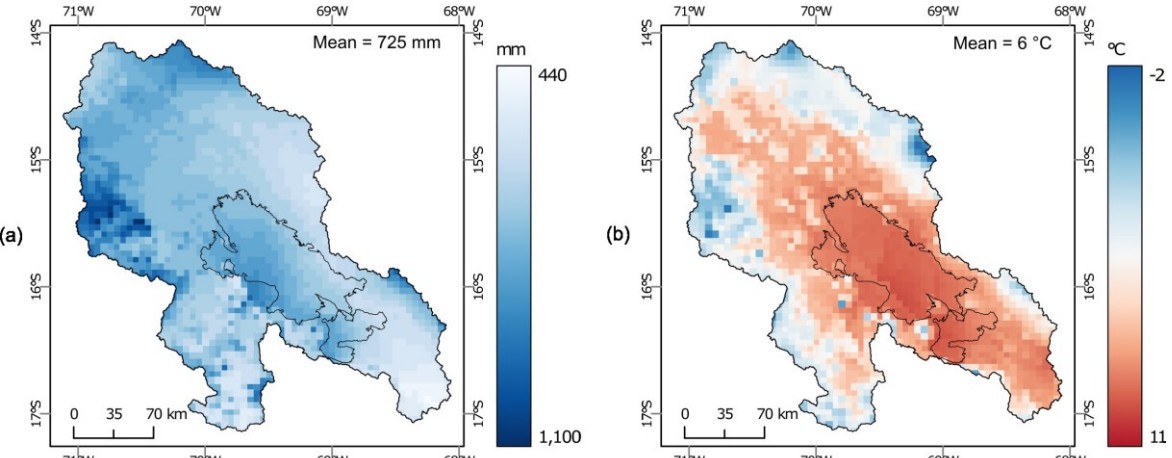

**Figure 2. Spatial distribution of (a) annual precipitation and (b) mean temperature for the hydrological period 1980–2016 according to GMET (Gridded Meteorological Ensemble Tool).**

## 2.3 Snow and glacierized areas

According to MODIS snow cover (Hall et al., 2002) computed over the period 2000–2016 based on a method described in Ruelland (2020), 80% of the upstream catchments are completely free of snow throughout the year. Areas where the snow cover persists for more than 20% of the year are located above 4,700 m a.s.l. These high elevations areas represent less than 0.5% of the total surface area. Glacierized areas are located above 4,600 m a.s.l. and represented 231 km$^2$ in the early 2000s (RGI Consortium, 2017), i.e. less than 0.5% of the total area. The estimated glacier volume is ~12 km$^3$ (Farinotti et al., 2019), which represents ~1.3% of the mean volume of Lake Titicaca (958 km$^3$). As a result, the contribution of snow and ice processes to streamflow (and lake dynamics) might be low in the catchments that contribute to the lake. However, this must be demonstrated by the water balance of the hydrosystem.

## 2.4 Croplands and irrigable area

According to the 2010 land cover map (Ministerio de Desarrollo Rural y Tierras, 2010; Ministerio del Ambiente, 2015) (see Fig. 1), cropland in the catchments covers 8,069 km$^2$, representing 17% of the total area in the upstream catchments. Agriculture is largely traditional, rainfed, and constrained by frequent droughts, periods of frost, and hailstorms (Garcia et al., 2007). The main crops are forage grasses, potatoes, grain barley, and quinoa (INTECSA et al., 1993c). A significant portion of the arable land is not cultivated due to crop rotation practices and agroclimatic constraints. For example, the ratio of cultivated land to arable land does not exceed 40% (INTECSA et al., 1993c). In terms of cropping calendar, potatoes are planted between October and November and their growing cycle lasts approximately six months until harvest. Quinoa is planted in September and their cycle is similar to that of potatoes. Some crops, such as beans and onions, are mostly irrigated and are planted between July and September. A crop coefficient ($Kc$) was generated using information on the cropping calendar and crop type available for some of the catchments (Autoridad Nacional del Agua, 2009, 2010; Instituto Nacional de



Estadística, 2015; Instituto Nacional de Recursos Naturales, 2008). Figure 3a shows the seasonal cycle of $Kc$ for the main crops. Most agriculture takes place between October and April. Alfalfa is cropped almost continuously throughout the cropping period.

The land cover maps of Peru and Bolivia (Ministerio de Desarrollo Rural y Tierras, 2010; Ministerio del Ambiente, 2015) do not distinguish between rainfed and irrigated areas (see Fig. 1). This is an important limitation to identifying changes in

irrigated areas over time. The only information available is the inventory of irrigation systems. The inventory on the Bolivian side was carried out in 2012 (Ministerio de Medio Ambiente y Agua, 2012). For the Peruvian side, the "rights of use" granted by the Autoridad Nacional del Agua (ANA) until 2023 were used (https://snirh.ana.gob.pe). The inventories provide information on latitude, longitude, irrigable area, and volume granted (Peru) or reference volume of irrigation (Bolivia). Figure 3b illustrates the available information on irrigation systems in terms of location and irrigable area. The irrigable area is 767

km$^2$ (see Table 1), which represents 1.6% of the upstream catchments that contribute to Lake Titicaca. Only 9.5% of the croplands are located in 'irrigable areas', i.e., cropland within the area of influence of an irrigation system that can potentially be irrigated. However, not all of the irrigable area is irrigated because irrigation depends on the availability of water in space and over time. Due to the lack of information, we assumed the irrigable area was constant over the period 1980–2016. Figure 3b also shows that most of the irrigation systems cover an area of less than 5 ha, i.e. small-scale irrigation predominates. Also,

irrigation is mostly practiced by smallholder farmers. Furrow irrigation is the most common system and irrigation network efficiency is about 35% (Autoridad Nacional del Agua, 2009; Instituto Nacional de Recursos Naturales, 2008; Ministerio de Medio Ambiente y Agua, 2012). The main sources of water are rivers and reservoirs (see Fig. 3) located in the upper parts of the upstream catchments. The main reservoir is Lagunillas (see Fig. 3b), built in 1995 for irrigation and drinking water, with a capacity of 500 million m$^3$. The remaining 15 reservoirs have capacities of less than 30 million m$^3$. Due to lack of data on

dam management, streamflow regulation from these dams was not accounted for in this study and we assumed that streamflow regulation has a limited impact on natural flows.



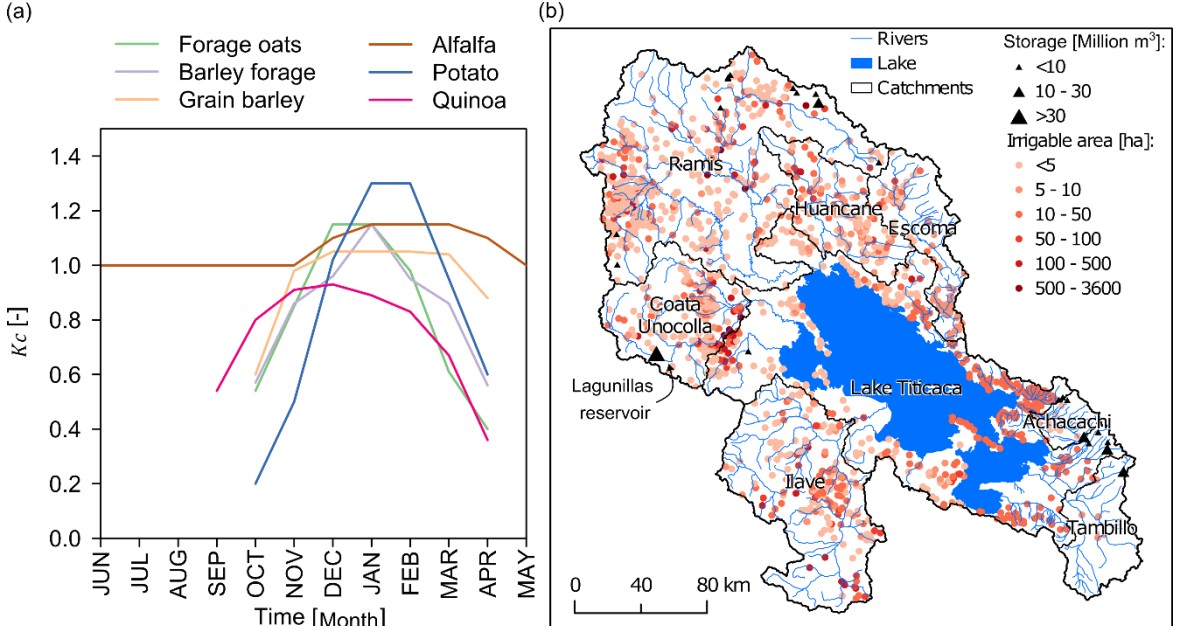

**Figure 3. Agricultural features in catchments located upstream of Lake Titicaca. (a) *Kc* showing the phenological cycle of the main crops (Instituto Nacional de Recursos Naturales, 2008; Autoridad Nacional del Agua, 2009, 2010; Instituto Nacional de Estadística, 2015). (b) location of irrigation systems in terms of irrigable area (Ministerio de Medio Ambiente y Agua, 2012; https://snirh.ana.gob.pe).**

## 2.5 Glaciological and hydrological control data

### 2.5.1 Geodetic mass balance of the glaciers

Within the study area, no observations are available at the scale of small glacierized catchments. However, since the 1990s, French and Bolivian glaciologists have been taking measurements on the Zongo catchment, which is the glacier-dominated catchment adjacent to our study area (see Fig. 1). Streamflows are measured at the Tubo gauge for a catchment of 3.5 km$^2$ of which 52% is glacierized area. Some data are missing in the records (see Fig. 6a). The annual mass balance estimated by the glaciological method (based on direct sampling of changes in the mass of the glacier) was also available for the hydrological period 1991–2016 (see Fig. 6b). Only geodetic mass balance data are computed at the scale of the entire Lake Titicaca hydrosystem (e.g. Dussaillant et al., 2019; Hugonnet et al., 2021). The Hugonnet et al. (2021) dataset, which is based on ASTER satellite stereo imagery, is available for the period 2000–2019. The mass balance at a 5-year time step is provided for RGI 6.0 glacier outlines. These data are not strictly direct observations and therefore include inaccuracies. Both data include the error associated with the mass balance of each glacier. The error range of the Hugonnet et al. (2021) dataset is smaller than the Dussaillant et al. (2019) dataset, and the interpolation of glacier elevation changes is based on Gaussian process regressions. For this reason, we used the Hugonnet et al. (2021) dataset.



### 2.5.2 Streamflow records

Seven streamflow gauges (see Table 1 and Fig. 1) with daily records were used in this study. The gauged (ungauged) area represents 76% (24%) of the total area of the catchments that feed the lake. The quality of the Peruvian gauge data can be considered satisfactory since monthly streamflow gauging is performed to calibrate the rating curves, but on the Bolivian side,

the quality of the Escoma and Achacachi data is questionable. According to SENAMHI–Bolivia, the river stage measured in Escoma and Achacachi are prone to systematic errors due to erroneous measurements made by the observers with limited measurement training (Escoma), and/or due to changes in the geomorphology of the riverbed (Achacachi). Streamflow gauging is only carried out twice in a hydrological year.

**Table 1. Main characteristics of the gauged and ungauged catchments that contribute to the lake. Glacier area was estimated using**
**RGI 6.0 glacier outlines (RGI Consortium, 2017). Cropland area (including rainfed and irrigated area) was estimated using the 2010 land cover maps of Peru and Bolivia (Ministerio de Desarrollo Rural y Tierras, 2010; Ministerio del Ambiente, 2015). Irrigable area was estimated using data from the inventories of agricultural land use rights (Peru) (https://snirh.ana.gob.pe), and irrigation systems (Bolivia) (Ministerio de Medio Ambiente y Agua, 2012). The reference year for irrigable area in Bolivia is 2012, and in Peru it has been updated to 2023. Elevations were extracted from a digital elevation model (DEM) at 90m spatial resolution from the Shuttle**
**Radar Topographic Mission (SRTM, Jarvis et al., 2008).**

| River | Source | Area [km$^2$] | Glacier area in 2000 [km$^2$] | Cropland in 2010 [km$^2$] | Irrigable area [km$^2$] | Elevation [m a.s.l.] Min | Elevation [m a.s.l.] Max |
|---|---|---|---|---|---|---|---|
| | Streamflow gauges | | | | | | |
| Ramis | Ramis | SENAMHI–Peru | 14,943 | 19 | 2,680 | 150 | 3,812 | 5,735 |
| Ilave | Ilave | SENAMHI–Peru | 7,814 | 0 | 262 | 39 | 3,813 | 5,587 |
| Coata Unocolla | Coata Unocolla | SENAMHI–Peru | 4,475 | 0 | 261 | 113 | 3,813 | 5,447 |
| Huancane | Huancane | SENAMHI–Peru | 3,518 | 0 | 333 | 19 | 3,814 | 5,079 |
| Suchez | Escoma | SENAMHI–Bolivia | 2,933 | 101 | 68 | 19 | 3,819 | 5,939 |
| Katari | Tambillo | SENAMHI–Bolivia | 2,612 | 3 | 255 | 31 | 3,832 | 5,905 |
| Keka | Achacachi | SENAMHI–Bolivia | 802 | 53 | 70 | 68 | 3,835 | 6,024 |
| Ungauged catchments | - | | 11,680 | 54 | 4,140 | 328 | 3,812 | 6,300 |

### 2.5.3 Lake water levels

We had access to data recorded at two water level gauges: Puno and Huatajata (see Fig. 1). The Puno gauge (also known as Muelle ENAFER) is managed by SENAMHI, Peru while the Huatajata gauge is managed by SENAMHI, Bolivia. The daily historical water levels from Puno are more reliable. In the case of Huatajata, inconsistencies were detected in the records made

prior to 1998. Additionally, during a field visit, it was observed that the Huatajata measurement scale is prone to displacement. Therefore, in this study we used data from Puno, which provides continuous daily water levels (in m a.s.l.) over the period 1982–2016.



## 3 Method

### 3.1 Modeling chain used to quantify the water balance in a high mountain lake hydrosystem

The water balance of Lake Titicaca was modeled at a daily time step for the hydrological period from September 1, 1981 to August 31, 2016 (hereafter 1982–2016). The water balance was modeled in a lumped mode store following the equation:

$$P_{lake} + Q_{in} - E_{lake} - Q_{out} = \frac{dh}{dt} \pm Q_{gw} \pm \varepsilon \tag{1}$$

where $P_{lake}$, $Q_{in}$, $E_{lake}$ and $Q_{out}$ are, respectively, direct precipitation over the lake, inflow from upstream catchments, evaporation from the lake, and downstream outflow. The term $dh/dt$ represents the storage change in the lake over a time

window. $Q_{gw}$ represents net groundwater exchange and $\varepsilon$ represents the error that cannot be explained by the components of the water balance. The unit of the water balance terms is mm d$^{-1}$.

The WEAP platform (Yates et al., 2005) was adapted and used to represent the water balance dynamics. WEAP offers several ways to simulate natural and influenced flows by considering water allocation in the context of water resources management. The models in WEAP typically seek a compromise between data availability and the complex representation of hydrologic

processes. This is essential in the context of poorly-gauged regions, where it is not possible to represent all hydrological processes in sufficient detail. Figure 4 shows the main processes and parameters of the modeling chain used. The approach used to estimate the components of the water balance with the proposed modeling chain are described in the following sections.

### 3.1.1 Direct precipitation over the lake

$P_{lake}$ was extracted from GMET for the outline of the lake. It is well known that large lakes have both a local and regional

influence on precipitation (Scott and Huff, 1996). The large area and volume of Lake Titicaca favor absorption of solar radiation and results in higher water temperatures than the surrounding area, which, in turn, induce convection and higher precipitation over the center of the lake (Roche et al., 1992). However, the magnitude and spatial distribution of precipitation over the lake are not well understood. GMET included two precipitation gauges located on two different islands in the lake. In this study, the extracted data were used directly without correcting for scaling factors. Significant underestimation of

precipitation could lead to a significant error.

### 3.1.2 Upstream inflow

$Q_{in}$ was estimated using a conceptual modeling approach that considers natural hydrological processes and irrigation in the upstream catchments. The model was applied using the same 100-m elevation bands in each catchment to account for snow and ice accumulation and melt, and glacierized and non-glacierized areas were distinguished in each elevation band. To

estimate the contribution of snow and ice to streamflow, a different degree-day model from the one available in WEAP was developed. The soil moisture model (SMM, part of WEAP) (Yates et al., 2005) that includes an irrigation module was used to simulate production and routing processes in the catchments.





 *(figure)*

**Figure 4. Main processes and parameters (in red) of the modeling chain used to simulate the water balance of Lake Titicaca.** $P$, $T$, $ETo$, $Pe$, $P_{lake}$, $Q_{in}$, $E_{lake}$, $Q_{gw}$, $Q_{out}$, and $h$ **stand for, respectively, precipitation, air temperature, reference evapotranspiration, effective precipitation, direct precipitation over the lake, upstream catchment inflow, evaporation from the lake, net groundwater exchange, downstream outflow and lake storage. Root zone and deep zone stores were modified based on Yates et al. (2005).**



For snow and ice processes, a degree-day model was applied that considered two stores: one for ice and one for snow (see Fig. 4) in a semi-distributed mode with 100m elevation bands. However, each glacier in each catchment was simulated separately.

For snow accumulation, the liquid ($P_{rain}$, in mm) and solid ($P_{snow}$, in mm) fractions of total precipitation were estimated from a linear separation between the snow ($T_s$) and rain ($T_l$) temperature thresholds according to values (see Table 2) recommended in Ruelland (2023). Potential (maximum) snowmelt was calculated as $DDF_{snow}(T - T_m)$, where $DDF_{snow}$ is the degree-day factor in mm day$^{-1}$°C$^{-1}$, $T$ is the air temperature in °C, and $T_m$ is the melting temperature threshold in °C. $T_m$ was calibrated according to two values ($T_{m,max}$ and $T_{m,min}$), where the maximum value occurs in austral summer and the minimum value in

winter. In outer tropical regions, the amplitude of the diurnal range of air temperature is indeed considerable in winter. This means that it is warmer during the daytime, which can increase melting, while cold conditions prevail at night. The seasonal variation of $T_m$ was calculated using the following equation:

$$T_m = \frac{T_{m,max} + T_{m,min}}{2} + \frac{T_{m,max} - T_{m,min}}{2} \sin\left(2\pi \frac{D+81}{365}\right) \tag{2}$$

where $T_{m,max}$ and $T_{m,min}$ are the maximum and minimum temperature thresholds in °C, and $D$ is the Julian day. The maximum

value of $T_m$ was assumed to occur on January 10 and the minimum on July 12.

Actual snowmelt ($M_{snow}$) was determined as a function of maximum snowmelt and snow accumulation. For ice melt ($M_{ice}$), the same approach was used as for potential snowmelt, except that $DDF_{snow}$ was replaced by an ice degree-day factor ($DDF_{ice}$). Ice melts when it is not covered by the snowpack. The daily mass balance ($B$, in mm w.e.) and effective precipitation in glacierized areas ($Pe_g$, in mm w.e.) in each elevation band ($j$) were computed as follows:

$$B_j = P_{snow,j} - M_{snow,j} - M_{ice,j} \tag{3}$$

$$Pe_{g,j} = P_{rain,j} + M_{snow,j} + M_{ice,j} \tag{4}$$

The annual mass balance in each evaluation band, $B_{a,j}$, was estimated as the sum of the daily mass balance in a hydrological year. The annual mass balance was also calculated for individual glaciers ($B_{a,g}$) for comparison with the available glaciological and geodetic mass balance. $B_{a,g}$ was calculated as:

$$B_{a,g} = \frac{\sum_{j=1}^{n}(B_{a,j} \times A_{g,j})}{A_g} \tag{5}$$

where $A_{g,j}$ and $A_g$ are glacier area in the elevation band $j$ and total glacier area, respectively.

The glacierized surface area (RGI Consortium, 2017) was fixed for the period simulated. The area provided for the year 2000 was considered as an intermediate value for the period 1982–2016. Ice thickness was also assumed to be infinite. The air temperature in each elevation band ($T_j$) was estimated as $T_{GMET,j} + \Gamma(Z_{GMET} - Z_j)$, where $T_{GMET,j}$ is the air temperature

derived from GMET for each elevation band, $Z_{GMET}$ is the mean areal elevation signal from GMET in the elevation band $j$, $Z_j$



is the mean elevation of the elevation band, and Γ is a constant temperature lapse rate that was set to the value calculated from GMET (i.e. 5.8°C km⁻¹). The precipitation extracted from GMET for each elevation band was used directly with no modification.

SMM (formerly WatBal) (Yates et al., 2005; Yates, 1996) was used for production and transfer processes, including irrigation

net consumption. SMM is a one-dimensional model based on two stores (see Fig. 4). It uses a differential equation and empirical functions to quantify evapotranspiration, surface runoff, interflow, deep percolation, and subsurface runoff. The first store represents the root zone and the second the deep zone (Yates et al., 2005). Because of its differential approach, the model can be run at different time steps (Yates, 1996). However, when SMM is used in relatively large catchments at fine time steps (e.g., daily), it may be necessary to consider runoff routing. The model without irrigation has seven free parameters as shown

in Figure 4, of which $Kc$ can be set using reference values from the literature (see Fig. 3a). In addition, there are two parameters associated with the initial state of the two stores called $z_1$ and $z_2$. Since this study used a daily time step and catchments are up to 15,000 km² in size, an additional store was considered for runoff routing with the Muskingum equation. SMM is driven by precipitation and reference evapotranspiration estimated by the modified Penman-Monteith method (Maidment, 1993) for a grass crop 0.12 m in height and with a surface resistance of 69 s m⁻¹. The climate input data are detailed in Sect. 2.2. The

effective precipitation in the elevation band $j$ of both the non-glacierized and glacierized fractions is given as:

$$Pe_j = (P_{rain,ng,j} + M_{snow,ng,j})A_{ng,j} + Pe_{g,j}(1 - A_{ng,j}) \qquad (6)$$

where $P_{rain,ng}$ and $M_{snow,ng}$ refer to rainfall and snowmelt in the non-glacierized fraction in mm. The term $A_{ng}$ is the relative area of the non-glacierized fraction.

In SMM, water requirements ($WR$) for irrigation are a function of crop evapotranspiration (from seasonal $Kc$ and reference

evapotranspiration) and the depletion of available water in the root zone store (see Fig. 4). The lower and upper irrigation threshold parameters ($L_{irr}$ and $U_{irr}$, see Table 2 and Fig. 4) dictate both the timing and quantity of water used for irrigation (Yates et al., 2005). When the relative soil moisture of the root zone store drops below the lower threshold, a water requirement is triggered and irrigation is supposed to be applied up to the upper threshold (Yates et al., 2005). The irrigation use of runoff ($IUR$) method was used to allocate water. This method consists of setting or calibrating a percentage ($IUR$) of a catchment's

runoff (before the runoff reaches the main river) that can be used for internal irrigation. $IUR$ focuses on water allocation at the catchment scale, especially when hundreds of irrigation systems are to be represented together. Simulating each of the irrigation systems shown in Figure 3 individually would not be feasible at the scale of this study, the $IUR$ approach is thus better suited. The irrigation net consumption, $IRR_{net}$, was calculated as:

$$IRR_{net} = min(Q_{wi} \times IUR, WR) \times (1 - IRR_{rf}) \qquad (7)$$

where $Q_{wi}$ is runoff without irrigation in mm, $IUR$ is a calibration parameter expressed as a percentage, $WR$ is the irrigation water requirement in mm, and $IRR_{rf}$ is the irrigation runoff fraction expressed as a percentage. The term $min(Q_{wi} \times IUR, WR)$ is the water withdrawn for irrigation. $IRR_{rf}$ is calculated as: (i) in the first iteration, SMM simulates



$Q_{wi}$; (ii) in the second iteration, runoff is simulated assuming that the full $WR$ is supplied; and (iii) finally, the $IRR_{rf}$ is estimated based on how much runoff would flow due to irrigation alone.

### 3.1.3 Evaporation from the lake

$E_{lake}$ was estimated using the Penman method for open water (Penman, 1948). This method is justified because it requires fewer data than an energy balance approach but is not as simple as a temperature-based approach. The Penman method also attempts to incorporate the energy balance in a simplified manner and includes mass transfer. The equation is given as:

$$E_{lake} = \frac{\Delta}{\Delta+\gamma} \frac{R_n - G}{\lambda} + \frac{\gamma}{\Delta+\gamma} f(U_2)(e_s - e_a) \tag{8}$$

where $E_{lake}$ is the evaporation in mm d$^{-1}$, $\Delta$ is the slope of the vapor pressure curve in kPa °C$^{-1}$, $\lambda$ is the latent heat vaporization set at 2.45 MJ kg$^{-1}$, and $\gamma$ is the psychrometric constant kPa °C$^{-1}$. The terms $R_n$ and $G$ are net radiation at the water surface and heat storage changes in MJ m$^{-2}$ d$^{-1}$, respectively. $f(U_2)$ is the function of wind speed measured at 2 m above the lake surface that is equal to $c(a + bU_2)$, where the constant of $a = 10$, $b = 5.4$, and $c = 0.26$ for Lake Titicaca were taken from Delclaux et al. (2007). Also, $e_s$ is the vapor pressure at the evaporating surface in kPa, and $e_a$ is the vapor pressure at 2 m above the lake surface in kPa. $\Delta$ is given as $(e_s - e_a)/(T_w - T)$, where $T_w$ and $T$ are evaporating surface temperature and air temperature in °C, respectively.

$R_n$ is the sum of net shortwave ($K$) and net longwave radiation ($L$). $K$ is given as $K_{in}(1 - \alpha)$, where $K_{in}$ is the incident solar radiation (MJ m$^{-2}$ d$^{-1}$) and $\alpha$ is the albedo of the water surface. The $L$ component is the difference between the incident flux ($L_{in}$) emitted by the atmosphere and clouds and outgoing radiation ($L_{out}$) from the evaporating surface. $L_{in}$ and $L_{out}$ can be estimated with Eq. (9) and Eq. (10), respectively. For $L_{in}$, we used the equation calibrated by Sicart et al. (2010) on the Zongo Glacier which is located at a distance of about 100 km from Lake Titicaca. The authors suggest that the calibration can be used in the central Andes.

$$L_{in} = C \left(\frac{e_a}{T+273.15}\right)^{1/7} (1.67 - \tau_{atm} 0.83)\sigma(T + 273.15)^4 \tag{9}$$

$$L_{out} = \varepsilon_w \sigma(T_w + 273.15)^4 \tag{10}$$

where for a daily time step, $C$ is equal to 1.24, $e_a$ is the vapor pressure in hPa, $\sigma$ is the Stefan–Boltzmann constant (set at 4.90×10$^{-9}$ MJ m$^{-2}$ K$^{-4}$ d$^{-1}$), $\tau_{atm}$ is the atmospheric transmissivity (-) that can be approximated as $K_{in}/S_{extra}$, $S_{extra}$ is theoretical shortwave irradiance (MJ m$^{-2}$ d$^{-1}$) at the top of the atmosphere, and $\varepsilon_w$ is water emissivity set at 0.98 (-).

The term $G$ (heat storage changes) in Eq. (8) was estimated using the approach and assumptions of the study conducted by Pillco Zolá et al. (2019) on Lake Titicaca.

$$G = c_w \rho_w \frac{V_{mix}}{A_{lake}} \frac{dT_w}{dt} \tag{11}$$




where $c_w$ is the specific heat of water (4.18×10⁻³ MJ kg⁻¹ °C⁻¹), $\rho_w$ is the water density (1000 kg m⁻³), $V_{mix}$ is the volume above the mixing depth in m³ and $A_{lake}$ is the surface area of the lake in m². $dT_w/dt$ is the change in water temperature (°C) over the time interval (day).

In terms of input data, air temperature was obtained from GMET, while all the other meteorological variables were obtained
from ERA5-Land (see Sect. 2.2). Lake surface water temperature (LSWT) time series are difficult to obtain and are not available for Lake Titicaca. A conceptual lumped model called Air2Water was thus used to estimate LSWT from air temperature (Piccolroaz et al., 2013; Toffolon et al., 2014). The Air2wateR 2.0.0 version available in R was used. ARC-Lake V3 data (MacCallum and Merchant, 2012) available for the period 1995–2012 were used to calibrate and evaluate the Air2wateR model. The spatial resolution of ARC-Lake V3 is 0.05° and the temporal resolution is daily. The model was
calibrated for the period 1995–2003 and evaluated using the period 2004–2012 (see Appendix B1). The performance of Air2Water in the independent control period was acceptable and captured seasonal and interannual variations very well. LSWT of ARC-Lake V3 had previously been compared with data measured at a buoy located in Lake Titicaca that recorded LSWT between 2019 and 2023. The comparison was made between the cell intersecting the measurement point and compared the range of fluctuations and seasonality, since there is no temporal link between the two sources of data. Measured LSWT and
LSWT obtained from ARC-Lake V3 fluctuated in a similar range and showed the same seasonal cycle (see Appendix B2).

### 3.1.4 Downstream outflow

$Q_{out}$ was simulated using the rating curve shown in Figure 5a. This curve was established 30 years ago based on a hydrodynamic simulation of the Desaguadero river (INTECSA et al., 1993b). The elevation corresponds to the vertical datum of Peru. The rating curve was used for the entire study period. The rating curve was implemented in the model in combination
with lake bathymetry (see Fig. 5b) carried out between 2016 and 2019 by the ALT.

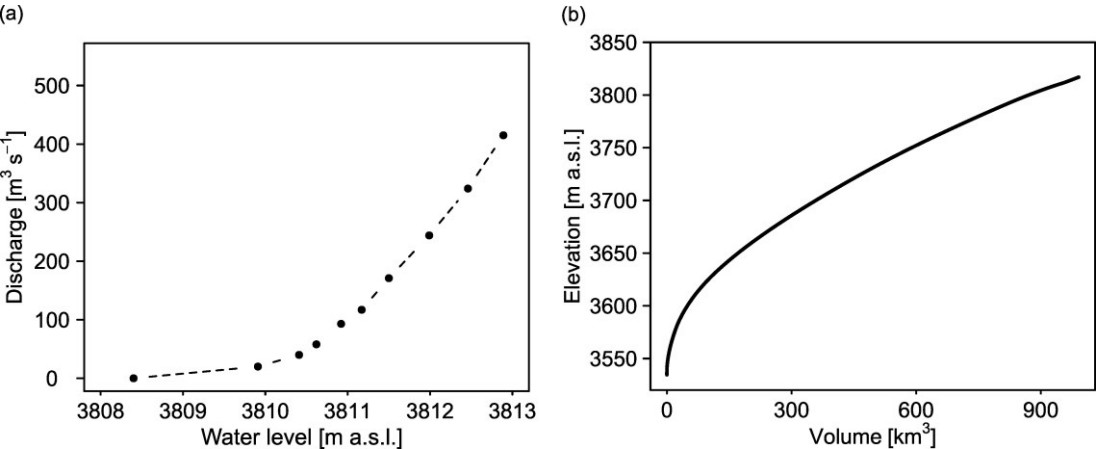

**Figure 5. Information used to simulate $Q_{out}$ for (a) the rating curve and (b) bathymetry. The rating curve of the lake outlet to the Desaguadero river was obtained from the master plan (INTECSA et al., 1993b). In both figures the elevation is referenced to the Peruvian vertical datum. The bathymetry (i.e. the relationship between lake water level and storage volume) was recorded by ALT**
**between 2016 and 2019.**





### 3.1.5 Storage change

$dh/dt$ was calculated directly from the water levels measured in Puno gauge. Storage change is basically the difference between the water level of a current time step and the previous time step.

### 3.1.6 Net groundwater exchange

$Q_{gw}$ was considered negligible. According to INTECSA et al. (1993a) the leakage from Lake Titicaca to the aquifers is very limited and the lake can be considered as an almost completely closed surface system. This is because the lake bed is composed of sediments with very low permeability. In this case, the only areas of high permeability would be limited to alluvial deposits saturated by water that flows mostly towards the lake. According to the same study, the inputs from alluvial deposits were 0.56 m$^3$ s$^{-1}$. Therefore, omitting $Q_{gw}$ from the water balance is justified.

### 3.2 Assessment of the modeling chain

The performance of the lake water balance model was evaluated using both the error term of Eq. (1) and root mean square error ($RMSE$) computed between observed and simulated water levels. Since the $Q_{in}$ and $E_{lake}$ terms of the lake water balance were estimated with models, intermediate calibration and evaluation were necessary. Evaporation measurements are not available in the study area to evaluate the performance of the $E_{lake}$. Therefore, a formal calibration and evaluation

procedure was implemented only for the $Q_{in}$ estimates. The procedure was applied sequentially, first to obtain the model parameters simulating snow and ice processes, and then to calibrate and evaluate the production and transfer model parameters at the scale of the catchments that contribute to the lake. The procedure used is described below.

### 3.2.1 Parameters of the snow and ice model

The approach used was to obtain the snow and ice model parameters (see parameters in Fig.4 and Table 2) in the Zongo

catchment and then transfer them directly to the Titicaca modeling chain. Firstly, the catchment model parameters were calibrated against the streamflow of the Tubo gauge for the hydrological period 2000–2010. The previous two hydrological years were used as a warm-up period. The irrigation module was disabled because the non-glacierized area was completely dominated by rocks and no crops are cultivated at the high elevations of the Zongo catchment. The parameters with the best performance against observed streamflow were obtained from 10,000 simulations generated with random hypercube sampling

of the Monte Carlo approach. The range of parameters tested is shown in Table 2. The Nash-Sutcliffe efficiency index (Nash and Sutcliffe, 1970) calculated on the root-mean-square transformed streamflow ($NSE_{sqrt}$) was used as the objective function to select the best-performing parameter sets for the calibration period. $NSE_{sqrt}$ was selected because we were looking for good compromise between high and low flows. According to Oudin et al. (2006) the $NSE_{sqrt}$ provides an intermediate fit of the unit hydrograph. Secondly, the model was evaluated against the mass balance observed from glaciological method (see Sect. 2.5.1.)



for the period 1992–2016 using the *RMSE* criterion. The evaluation was performed with other internal variables of the model, because the goal was to obtain the set of parameters associated with the snow and ice stores. Good internal consistency of the glacier model was thus desirable. In addition, the 25-year time window made it possible to evaluate the ability of the model to simulate the interannual and decadal variation in the mass balance of the Zongo glacier. Figure 6 shows the model's calibration (see Fig. 6a) and evaluation (see Fig. 6b) performances. Figure 6a shows that the model reproduced streamflow reasonably

well. Mass balance was also simulated reasonably well. The *RMSE* obtained (349 mm) is in the range of the error of the mass balance based on the glaciological method (400 mm) (Sicart et al., 2007).

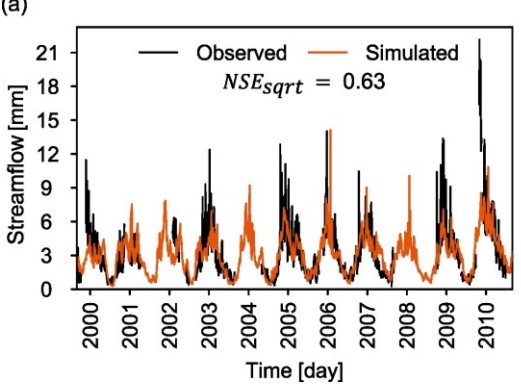
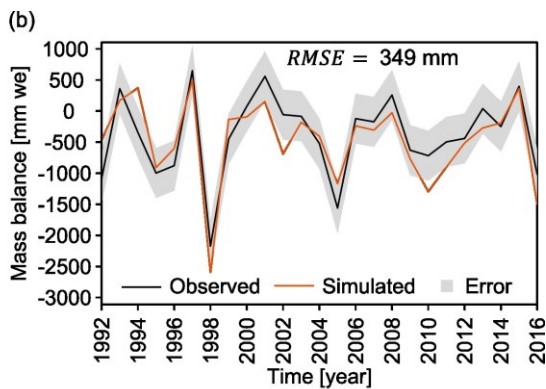

**Figure 6. Performance of the catchment model in the Zongo catchment versus (a) observed streamflow at the Tubo gauge, and (b) annual mass balance estimated using the glaciological method. The mass balance estimated using the glaciological method has an**
**error of 400 mm year$^{-1}$ (Sicart et al., 2007).**

**Table 2. Parameters of the model implemented in the Zongo catchment and their associated fixed values or ranges tested in order to obtain the parameters of the snow and ice model.**

| Parameter | Meaning | Unit | Fixed values or ranges tested |
|---|---|---|---|
| $T_s$ | Snow threshold temperature | °C | -1 |
| $T_l$ | Rain threshold temperature | °C | 3 |
| $T_{m,max}$ | Maximum melting temperature threshold | °C | [-0.5; 1] |
| $T_{m,min}$ | Minimum melting temperature threshold | °C | [-3; -0.5] |
| $DDF_{snow}$ | Snow degree-day factor | mm d$^{-1}$ °C$^{-1}$ | [1; 6] |
| $DDF_{ice}$ | Ice degree-day factor | mm d$^{-1}$ °C$^{-1}$ | [2; 15] |
| $Sw$ | Soil water capacity | mm | [150; 250] |
| $RRF$ | Runoff resistance factor | - | [4; 15] |
| $Ks$ | Root zone conductivity | mm d$^{-1}$ | [1; 6] |
| $PFD$ | Preferred flow direction | - | [0.3; 0.87] |
| $Dw$ | Deep water capacity | mm | [300; 600] |
| $Kd$ | Deep conductivity | mm d$^{-1}$ | [1; 3] |
| $Z_1$ | Initial condition | % | 30 |
| $Z_2$ | Initial condition | % | 30 |
| $L_{irr}$ | Lower threshold | % | 0 |
| $U_{irr}$ | Upper threshold | % | 0 |
| $IUR$ | Irrigation use of runoff | % | 0 |
| $k$ | Time travel | d | [0.5; 5] |
| $X$ | Diffusion | - | 0.2 |
| | | # of free parameters | 11 |



### 3.2.2 Upstream catchment model

The upstream catchment model shown in Figure 4 has 15 parameters. However, seven of the parameters were set to reduce the
number of free parameters. Four parameters ($T_{m,max}$, $T_{m,min}$, $DDF_{snow}$ and $DDF_{ice}$) related to snow and ice store were set to values obtained in the Zongo catchment. The simulated mass balance of all glaciers of the Titicaca hydrosystem was compared with the geodetic mass balance (Hugonnet et al., 2021) for the period 2000–2009. Similarly, two parameters of the irrigation module, $L_{irr}$ and $U_{irr}$, were set to 80%. Winter et al. (2017) used a threshold of 100% for furrow irrigation in California. However, a value of 80% is reasonable for our study area because it is irrigated in conditions of limited water availability. $X$
(routing store) was set to default value (0.2) in WEAP. A total of eight free parameters were kept, as shown in Table 3. The procedure used to obtain the set of parameters with the best performance and subsequent evaluation consisted of the four steps described below.

First (Step 1 in Fig. 7), the model was run for the period 1980–2016 (of which the first two years were used as a warm-up period) with 10,000 parameter sets generated from a random sample of hypercubes from the Monte Carlo approach within the
parameter intervals tested (see Table 3). Second (Step 2 in Fig. 7), the best performing parameter sets in terms of $NSE_{sqrt}$ were identified along with subperiods consisting of (i) 5 continuous years (i.e. seven subperiods between 1982 and 2016); and (ii) 5 discontinuous years identified as the coldest, warmest, driest, and wettest (i.e. four discontinuous subperiods between 1982 and 2016). The 11 best performing parameter sets were selected using different subperiods. Third (Step 3 in Fig. 7), the mean of 11 streamflow simulations generated with the selected parameter sets was calculated. Fourth (Step 4 in Fig. 7), the
performance of the mean of the streamflow simulations over the 11 subperiods was evaluated using the $NSE_{sqrt}$ criterion.

Two objectives justify the use of seven continuous and four discontinuous subperiods. The first objective was to evaluate the transferability of the model parameters to non-stationary conditions within the period 1982–2016, including particularly contrasted subperiods in terms of precipitation and temperature. In addition, continuous subperiods were suitable for assessing the transferability of the water allocation parameter ($IUR$). If there had been a significant and sustained increase in irrigation
withdrawals, the $IUR$ parameter would not be transferable over time and would therefore vary over the period 1982–2016. However, it is important to note that irrigable area, crop types, and $Kc$ remain under the assumption of stationarity. The second objective was to consider the parameterization uncertainty through uncertainty envelopes and confidence intervals in the simulations resulting from the 11 parameter sets used over the whole 1982–2016 period.



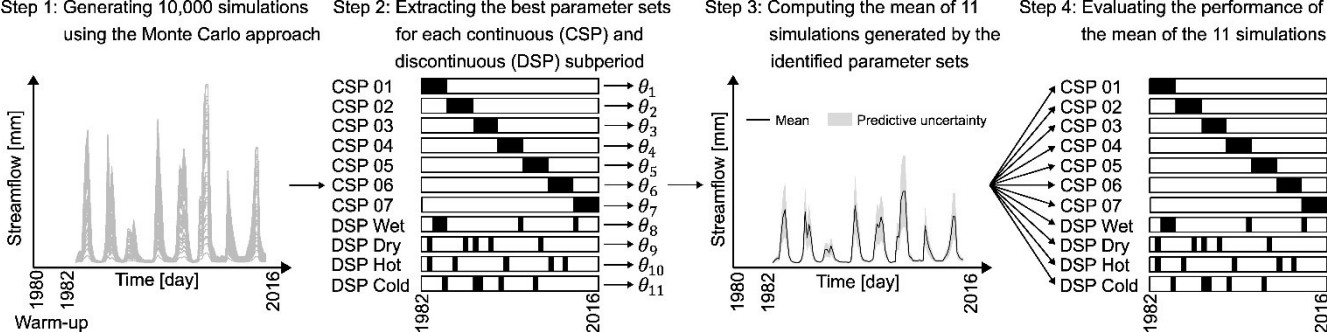

**Figure 7. Procedure used to evaluate the upstream catchment model to estimate the $Q_{in}$ in each gauged catchment.**

### 3.2.3 Sensitivity assessment of model predictability to irrigation and to snow and ice processes

To evaluate the sensitivity of the modeling chain to net irrigation consumption and to snow and ice processes, different processes were progressively excluded from the modeling structure. Three model structures were tested under the following configuration: (i) Model A, which is the full reference model structure (as in Fig. 4); (ii) Model B, where the processes associated with snow and ice processes (accumulation and ablation) were excluded but irrigation was maintained; and (iii) Model C, where both snow and ice processes and irrigation consumption were excluded. The objective was to evaluate whether a simpler structure in terms of the hydrological processes considered performed better or worse than a more complex structure in simulations of catchment streamflow and lake water levels. This is justified because, *a priori*, snowfall as well as the proportion of glacierized area and irrigable area are very limited in the catchments that contribute to the lake (see Table 1) and their impact on the streamflow and water level prediction may be negligible. In addition, the data used to represent and control these processes are very limited, which can lead to inaccuracies that could worsen the simulations of streamflow and lake water levels instead of improving them. Table 3 shows the active and inactive parameters in the three model structures.





**Table 3. Parameters of the upstream catchments model of Lake Titicaca and associated fixed values or ranges tested. The ranges presented were used to generate the random sample of hypercubes in the Monte Carlo approach.**

| Parameter | Name | Unit | Model A | Model B | Model C |
|---|---|---|---|---|---|
| $T_s$ | Snow threshold temperature | °C | -1 | - | - |
| $T_l$ | Rain threshold temperature | °C | 3 | - | - |
| $T_{m,max}$ | Melting temperature threshold | °C | -0.2 | - | - |
| $T_{m,min}$ | Melting temperature threshold | °C | -2.5 | - | - |
| $DDF_{snow}$ | Snow degree-day factor | mm d$^{-1}$ °C$^{-1}$ | 2.3 | - | - |
| $DDF_{ice}$ | Ice degree-day factor | mm d$^{-1}$ °C$^{-1}$ | 7.7 | - | - |
| $Sw$ | Soil water capacity | mm | [150; 250] | [150; 250] | [150; 250] |
| $RRF$ | Runoff resistance factor | - | [4; 15] | [4; 15] | [4; 15] |
| $Ks$ | Root zone conductivity | mm d$^{-1}$ | [1; 6] | [1; 6] | [1; 6] |
| $F$ | Preferred flow direction | - | [0.3; 0.87] | [0.3; 0.87] | [0.3; 0.87] |
| $Dw$ | Deep water capacity | mm | [300; 600] | [300; 600] | [300; 600] |
| $Kd$ | Deep conductivity | mm d$^{-1}$ | [1; 3] | [1; 3] | [1; 3] |
| $Z_1$ | Initial condition | % | 30 | 30 | 30 |
| $Z_2$ | Initial condition | % | 30 | 30 | 30 |
| $L_{irr}$ | Lower threshold | % | 80 | 80 | - |
| $U_{irr}$ | Upper threshold | % | 80 | 80 | - |
| $IUR$ | Irrigation use of runoff | % | [30; 80] | [30; 80] | - |
| $k$ | Time travel | d | [0.5; 5] | [0.5; 5] | [0.5; 5] |
| $X$ | Diffusion | - | 0.2 | 0.2 | 0.2 |
| | # of free parameters | | 8 | 8 | 7 |

### 3.2.4 Transferring parameters to the ungauged catchments

The approach used to transfer the parameters to the ungauged catchments consisted of the following steps: (i) the median of the parameter sets obtained for the seven gauged catchments was calculated for each subperiod, thus generating 11 parameter sets; (ii) the upstream catchment model was run for the 11 parameter sets; and (iii) the mean of the 11 streamflow simulations was calculated, including the confidence interval.

## 4 Results

### 4.1. Modeling chain performance

### 4.1.1 Performance of the geodetic mass balance simulated by the model for Titicaca glaciers

Figure 8 shows the comparison between the simulated and geodetic mass balance for the glaciers located in the different upstream catchments of Lake Titicaca. The scatter plots show significant variability in model performance, with some glaciers (represented by each point) distributed close to the identity line, while others deviate significantly from the identity line,
indicating variability in the simulation performance depending on the spatial distribution of glaciers. For this purpose, the scatter plots are presented as a function of the catchment (the color of each dot), the surface area of the glacier (see Fig. 8a), and the mean glacier elevation (see Fig. 8b). The model performed more effectively in the catchments that concentrate 92% of the glacierized area (Achacachi, Ungauged catchments, Tambillo and Escoma). The model performed very poorly in Ramis (see Table 4), but the proportion of glacierized area in that catchment is very small (0.1%). Therefore, the proportion is expected



to have a limited impact on streamflow predictions. Additionally, the model performs much better in the case of large glaciers than small glaciers (see Table 4). For example, for large glaciers representing 84% of the glacierized area, the weighted simulated mass balance for all catchments has a bias of 11%, which is within the error range of the geodetic mass balance (see Table 4). Much more obvious is the dependence on elevation (see Fig. 8b). The model performed very poorly on glaciers with mean elevations below 5,100 m a.s.l., which represent only 10% of the simulated glaciers. At elevations above 5,200 m a.s.l.,

i.e. 68% of the glaciers, the points are distributed around the identity line. This distribution could also be due to precipitation inaccuracies in some catchments. In this case, precipitation could be overestimated (underestimated) at higher (lower) elevations.

The transfer of the snow and ice parameters identified on the Zongo catchment to all the Titicaca catchments revealed interesting aspects, for example: (i) the limitations of applying the parameters to all glaciers, and (ii) an apparent dependency

on elevation, although the inaccuracies may also be associated with inaccuracies in precipitation and air temperature. Despite these issues, acceptable simulations were obtained for high-elevation and large-area glaciers. At the catchment scale, the underestimation and overestimation of glacier-wide mass balance (GMB) are compensated for, and the biases are relatively small. However, this is not the case in Ramis where the GMB of all the glaciers is overestimated. The sensitivity of streamflow prediction to snow and ice processes described in Sect. 4.1.2 below provide more clarity concerning the impact of accumulation

and ablation estimates.

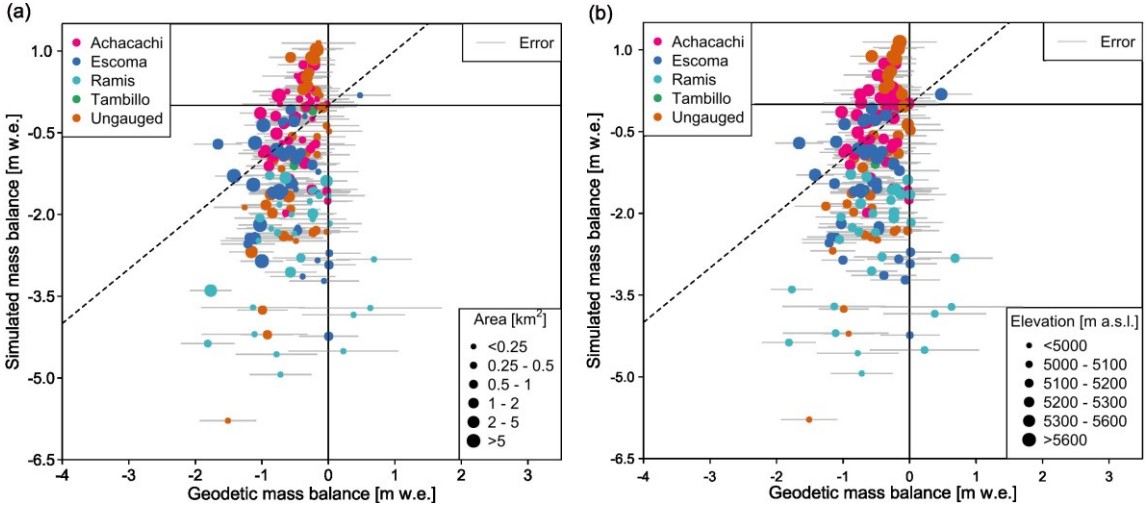

**Figure 8. Scatter plots comparing the simulated mass balance and the geodetic mass balance for the period 2000–2009. The simulations are compared with the geodetic mass balance of Hugonnet et al. (2021) where the size of the dots is a function of (a) glacier area and (b) mean glacier elevation. The identity line is indicated by the dashed lines. The geodetic mass balance error is**

**indicated by the gray line.**



**Table 4. Comparison of the simulated and geodetic mass balance (MB) in upstream catchments for the period for 2000–2009. The catchment scale mass balance was calculated as an area weighted average of each glacier. The error corresponds to the geodetic mass balance and was calculated as a weighted average for each catchment.**

| | Small glaciers (<1 km$^2$) | | | | Large glaciers (≥1 km$^2$) | | | |
| | | Simulated | Geodetic | | | Simulated | Geodetic | |
| | Area | MB | MB | Error | Area | MB | MB | Error |
| Catchment | [km$^2$] | [mm we] | [mm we] | [mm we] | [km$^2$] | [mm we] | [mm we] | [mm we] |
|---|---|---|---|---|---|---|---|---|
| Achacachi | 13 | -330 | -395 | 402 | 40 | -306 | -610 | 299 |
| Escoma | 6 | -1,973 | -453 | 428 | 95 | -1,246 | -929 | 313 |
| Ramis | 5 | -2,331 | -656 | 453 | 14 | -2,002 | -624 | 329 |
| Tambillo | 2 | -506 | -352 | 364 | 2 | -43 | -343 | 326 |
| Ungauged | 11 | -1,505 | -464 | 411 | 43 | -251 | -462 | 311 |
| All glaciers | 37 | -1,322 | -462 | 414 | 194 | -760 | -683 | 311 |

### 4.1.2 Performance and sensitivity of the upstream catchment model to irrigation and to snow and ice processes

Figure 9 shows the distribution of the performance of the three modeling options (Model A, Model B, and Model C) in the evaluation made for each catchment. A striking feature is that the performance in the catchments on the Peruvian side (Ramis, Ilave, Coata Unocolla and Huancane) was significantly better than in the catchments on the Bolivian side (Escoma, Tambillo and Achacachi). The performance of the three models is distributed symmetrically in the boxplots in each catchment. This suggests that there are no significant differences in performance between the three models. Therefore, $Q_{in}$ is not very sensitive to net irrigation consumption nor to snow and ice processes. This reflects the fact that the upstream catchments of Lake Titicaca are dominated by a pluvial monsoonal climate regime and that contributions from glacierized areas and snow have little influence on the $Q_{in}$ prediction. However, in terms of mean and median $NSE_{sqrt}$, Models A and B generally performed slightly better than Model C (from which both snow and ice as well as irrigation processes were excluded), but the differences are marginal, in most cases the difference is between 1% and 3%. However, in Achacachi, the $NSE_{sqrt}$ obtained with the full model A is 6% higher than the value obtained with models B and C. This suggests that accounting for processes associated with snow and ice accumulation and melt is relatively more important in catchments with more glacierized area. *A priori*, these results show that when irrigation and snow and ice processes are taken into consideration, $Q_{in}$ prediction only improves marginally. However, this must be confirmed by evaluating the sensitivity of the simulated water levels in Lake Titicaca to the different model structures (see Sect. 4.1.3.).





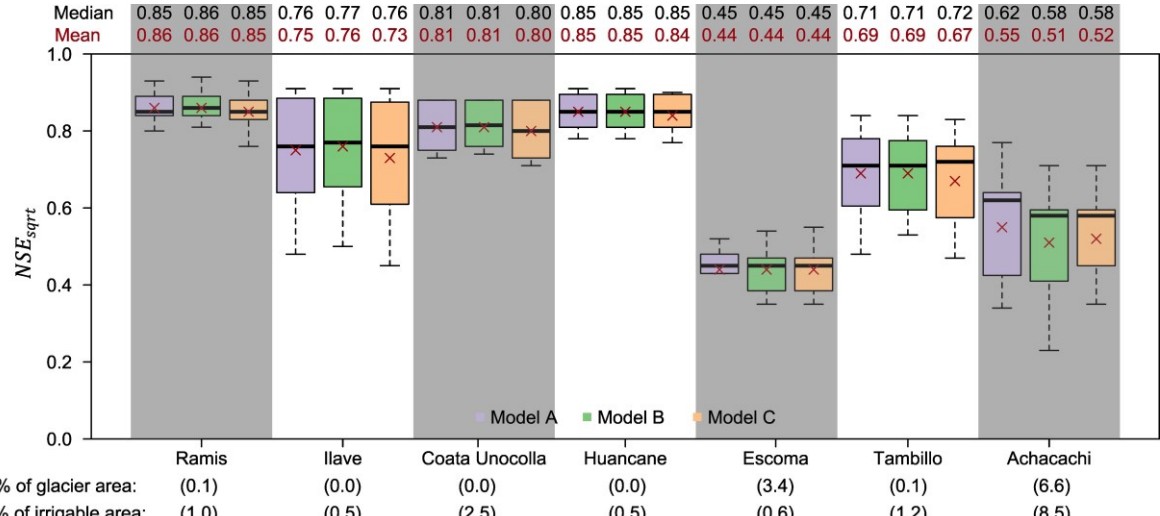

**Figure 9. Performance distributions of the three modeling options (A, B and C) in each gauged catchment in evaluation. The size of the sample in each boxplot is 11 (resulting from the procedure presented in Figure 7). Model A is the full reference model structure (see Fig. 4), Model B accounts for irrigation water consumption but not for snow and ice processes, and Model C includes neither snow and ice processes, nor consumption of water for irrigation.**

### 4.1.3 Performance of the modeling chain with respect to lake water levels

Figure 10 shows the simulation of daily water levels in Lake Titicaca over the 1982–2016 period. Three different estimates of $Q_{in}$, were used, one from Model A, one from B, and one from C. The results show that the models are able to simulate the amplitude and frequency of annual, interannual, and decadal water level fluctuations reasonably well. Visually, the performances of the three $Q_{in}$ estimates appear to be relatively similar. Based on the $\varepsilon$ term, Model B performed better than Models A and C. The difference in the $\varepsilon$ term between models A and B was 0.01 mm. When both snow and ice processes and net irrigation consumption were excluded, the error increased by 0.03 mm d$^{-1}$. However, the differences in error were marginal. Figure 10 shows that in the mostly dry years of the 1990s, Models A and B simulated daily water levels better. However, the performance measured by the *RMSE* differed in the error term. In that case, Model C performed best. This is because the *RMSE* was calculated using daily water level data (cumulative change in storage over time), whereas the $\varepsilon$ term was computed directly from the water balance at each time step. Inaccuracies in some time steps were then propagated to later time steps due to the slow response of the lake. The *RMSE* obtained was very small compared to the average water level (3,809.7 m a.s.l.).

A striking feature of Figure 10 is the systematic underestimation of daily water levels between 2001 and 2010. This is related to inaccuracies in the estimation of water balance terms. The hydrological response of Lake Titicaca is relatively slow, and it was possible to verify that significant errors were present between 2001 and 2004, but were not underestimated over the whole decade (2001–2010) (see Appendix C1). In 2001, the dam built to regulate outflows into the Desaguadero river was completed and dredging of the Desaguadero river had begun. This could mean fewer outflows into the Desaguadero and consequently more storage in the lake. However, even assuming there were no downstream outflows in those years would not compensate for the underestimation. Looking for other sources of error, discussions with the Lake Titicaca Authority (ALT) revealed that



the lake sometimes received inflows from the Desaguadero river, especially in wet years. This could be the case, since 2001 and 2004 were wet hydrological years, but other than verbal communication, there are no records to support such a claim. Concerning streamflow in the catchments that contribute to the lake, the results suggest that water level prediction is also not very sensitive to net irrigation consumption and snow and ice processes. Model A produced a reasonably realistic simulation

with an even smaller error, with marginal differences, than model C. Consequently, Model A (including irrigation and snow and ice processes) can be considered as a reference structure to provide an estimate of the water balance of the Lake Titicaca hydrosystem.

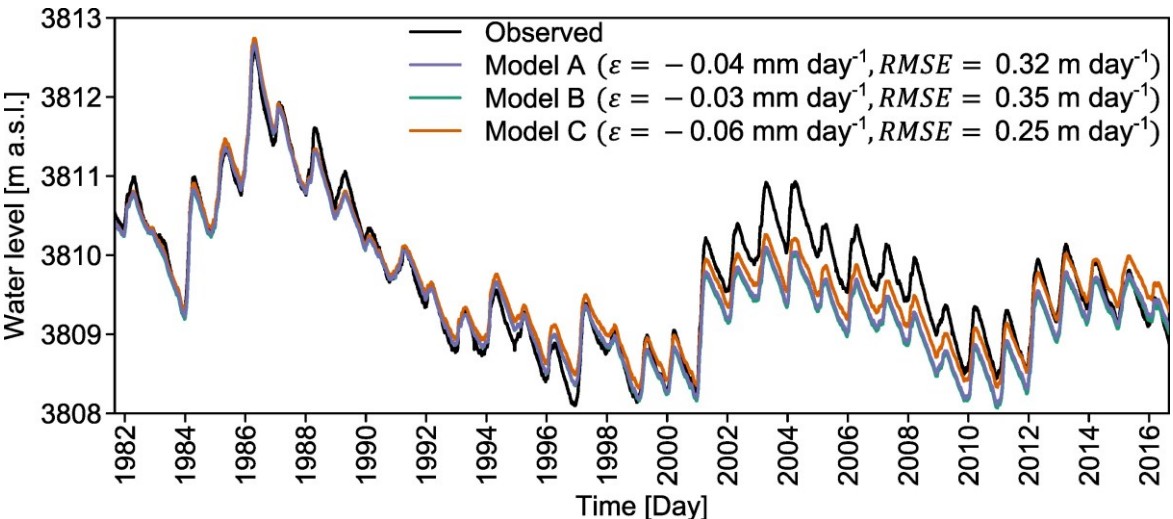

**Figure 10. Performance of the modeling chain compared with the water levels of Lake Titicaca measured in Puno. Modeling was**
**evaluated using the error ($\varepsilon$) term of the lake water balance and $RMSE$. Three simulations of water levels are presented because the models produced three different estimates of $Q_{in}$.**

## 4.2. Water balance simulated by the modeling chain

### 4.2.1. Simulated water balance in upstream catchments

Table 5 shows the mean annual water balance for the hydrological period 1982–2016 simulated by Model A. Some terms
($AE_{ni}$, $IRR$, and $Q_{in}$) presented in Table 5 were calculated as the average of 11 ensemble members. At the scale of all upstream catchments (Table 5), annual precipitation is 723 mm, of which 6% is estimated to be snowfall. Thus, despite the elevations at which the study area is located (>3,810 m a.s.l.), the precipitation regime is clearly dominated by rainfall. This regime has a unimodal pattern with a strong seasonal cycle between the rainy and dry seasons (see Fig. 11). This is very characteristic of the outer tropical areas, where precipitation occurs mainly in the summer and the dry season in winter. In the upstream
catchments, snowmelt accounts for 6% of total input ($P_{rain} + M_{snow} + M_{ice}$). Snow remains on the ground for only a very short time, as there is practically no time lag between snowfall and snowmelt (see Fig. 11). Snowfall and snowmelt have a less accentuated seasonal cycle, as the rare precipitation that occurs in the dry season is mostly snow. The contribution of ice melt was simulated to be 6 mm yr$^{-1}$ on average, which represents only 1% of the total input in upstream catchments. However, the





contribution of ice melt is slightly higher in the Achacachi and Escoma catchments, where the simulated value is 7%. Most ice
melt occurs between October and November (see Fig. 11) because the temperature is above the melting threshold and glaciers
are mostly free of snow cover. $\Delta_{ice}$ is negative, indicating a loss in ice stock. However, it is worth noting that the morphometric
change of the glacier was not dynamically simulated in this study. This limitation does not have a major impact on the water
balance of Lake Titicaca, as the glacier contributes very little to the streamflow at the catchment outlet.

Annual actual evapotranspiration ($AE$) was simulated at 570 mm of which 2% corresponds to net irrigation water consumption
($IRR$). $AE$ is highest between January and March, and lowest between July and
September (see Fig. 11). $IRR$ is concentrated in the transition season (see Fig. 11). The simulated streamflow is about 153 mm,
which represents 21% of the total outflow in upstream catchments ($AE + Q_{in}$), the remaining 79% being actual
evapotranspiration. $IRR$ represents 7% of $Q_{in}$. The seasonal cycle of $Q_{in}$ shows that the peak is in February, while $Q_{in}$ is low
from June to November. The predictive uncertainty associated with the upstream catchment model parameters is shown in Fig.
11 for the terms $AE$, $IRR$ and $Q_{in}$. The prediction range of $AE$ is very narrow (see Fig. 11) compared to that of $IRR$ and $Q_{in}$
indicating that evapotranspiration is less sensitive to the model parameters. The range of prediction of $IRR$ is quite wide (see
Fig. 11), indicating that the simulations are very sensitive to SMM parameters. The range of the prediction of $Q_{in}$, is also wide
(see Fig. 11). However, the average predictions fitted the measured streamflow and model performance reasonably well (see
Fig. 9).

**Table 5. Mean annual water balance [mm] in the upstream catchments for the hydrological period 1982–2016 simulated with Model A. $P$, $P_{rain}$, $P_{snow}$, $M_{snow}$, $M_{ice}$, $AE_{ni}$, $IRR$, $\Delta_{ice}$, $\Delta_{snow}$, $\Delta_{SM}$ and $Q_{in}$ represent total precipitation, rainfall, snowfall, snowmelt, ice melt, actual evapotranspiration in non-irrigated area, net consumption of irrigation water, variation in ice storage, variation in snow storage, variation in soil moisture storage, and streamflow in upstream catchments, respectively. The water balance follows the equation $Q_{in} = P - AE_{ni} - IRR - \Delta_{ice} - \Delta_{snow} - \Delta_{SM}$.**

| Catchment | $P$ | $P_{rain}$ | $P_{snow}$ | $M_{snow}$ | $M_{ice}$ | $AE_{ni}$ | $IRR$ | $\Delta_{ice}$ | $\Delta_{snow}$ | $\Delta_{SM}$ | $Q_{in}$ |
|---|---|---|---|---|---|---|---|---|---|---|---|
| Ramis | 777 | 726 | 51 | 51 | 3 | 610 | 7 | -3 | 0 | 2 | 162 |
| Ilave | 685 | 656 | 29 | 29 | 0 | 528 | 5 | 0 | 0 | 0 | 153 |
| Coata | 889 | 776 | 113 | 113 | 0 | 615 | 21 | 0 | 0 | 0 | 253 |
| Huancane | 664 | 647 | 17 | 17 | 0 | 498 | 5 | 0 | 0 | -1 | 163 |
| Escoma | 618 | 533 | 85 | 82 | 56 | 490 | 5 | -53 | 0 | 1 | 175 |
| Tambillo | 537 | 526 | 11 | 10 | 1 | 447 | 7 | 0 | 0 | 49 | 33 |
| Achacachi | 741 | 573 | 168 | 142 | 49 | 524 | 40 | -24 | 0 | 11 | 190 |
| Ungauged | 699 | 683 | 16 | 13 | 5 | 558 | 20 | -2 | 0 | 2 | 121 |
| All catchments | 723 | 679 | 44 | 43 | 6 | 559 | 11 | -5 | 0 | 5 | 153 |






**Figure 11. Seasonal cycle (monthly average for the period 1982–2016) of the water balance in the upstream catchments of Lake Titicaca simulated with Model A.** $P$, $P_{snow}$, $M_{snow}$, $M_{ice}$, $AE$, $IRR$, and $Qin$ represent, total precipitation, snowfall, snowmelt, ice melt, actual evapotranspiration, irrigation net water consumption, and streamflow in the upstream catchments, respectively. For some terms ($AE$, $IRR$, and $Qin$) the gray bars were estimated as the mean of the 11 ensemble members resulting from the procedure shown in Fig. 7. The predictive uncertainty is presented for both the entire prediction range (i.e. all predictive uncertainty) and for the 95% confidence interval. "All predictive uncertainty" was estimated as the maximum and minimum values of the ensemble members. The terms associated with snow and ice are not subject to predictive uncertainty because fixed parameters were used (see Table 3).





### 4.2.2 Simulated water balance in the lake

Figure 12a shows the annual evolution of the water balance terms of Lake Titicaca. Over the period 1982–2016, average annual precipitation over the lake was 744 mm (σ = 144 mm) and inflow from upstream catchments 958 mm (σ = 392 mm). This means that 44% (56%) of the inflows come from direct precipitation over the lake (comes from upstream watersheds). Regarding outflows, annual evaporation over the lake is 1,616 mm (σ = 28 mm) and the downstream outflow is 121 mm (σ = 191 mm), thus 93% of the losses are due to evaporation. The measured storage change for the period 1982–2016 was -50 mm, indicating a drop in water level. The simulated change in storage was -35 mm, which indicates an overestimation. Therefore, the water balance closure has an error of about -15 mm. Compared to evaporation, $P_{lake}$ and $Q_{in}$ showed significant interannual variability (see Fig. 12a). $E_{lake}$ was subject to less pronounced interannual variability, but showed an increasing trend over the period 1982–2016 due to the increase in temperature (about +0.1°C/decade). This means that the interannual variability of water levels depends to a large percent on variations in precipitation over the lake and in the upstream catchments. The highest values of both $P_{lake}$ and $Q_{in}$ occurred between 1985 and 1986 and caused large floods around the lake and in the Desaguadero river, where discharges reached about 900 mm yr$^{-1}$. In the 1990s, inflows were the lowest in the period studied, resulting in a mostly negative change in storage. Substantial inflows to the lake in the early 2000s led to a significant positive change in storage, although this was followed by another dry period that lasted until 2012.

Figure 12b shows a very marked seasonal cycle for precipitation and upstream inflow. For example, the monthly peak in upstream inflow is about 230 mm (in February), while in the dry season the values are very close to 15 mm (in September). One of the features is the lag of upstream inflow with respect to direct precipitation over the lake, evidence for the relatively slow hydrological response in the upstream catchments because of the size of the catchment area. The seasonal cycle of evaporation is less marked than that of precipitation and upstream inflow. Although air temperature shows strong seasonality, evaporation is also influenced by other meteorological variables, and heat storage plays a critical role in the seasonal cycle. The peak of the mean monthly evaporation is around 170 mm (in January), while the minimum value is around 95 mm (in August). In Figure 12b, it is also interesting to observe the seasonal cycle of the error term, which is mostly positive in the rainy season and mostly negative in the dry season. This could indicate that the lake receives net groundwater inflow during the rainy season and is subject to net outflow during the dry season. However, the magnitudes cannot be directly attributed to net groundwater flow because the error term includes both the uncertainty associated with estimating the other terms of the water balance ($P_{lake}$, $Q_{in}$, $E_{lake}$, and $Q_{out}$) and assumptions concerning the groundwater flow.



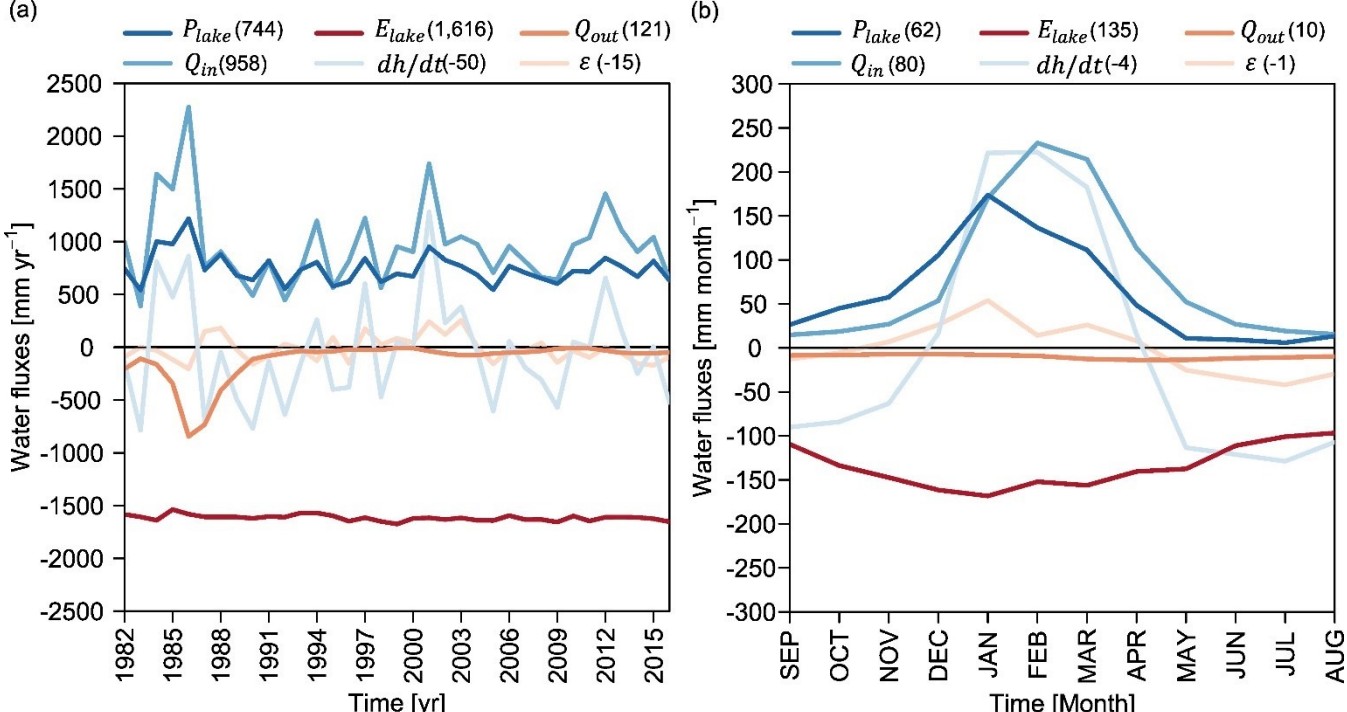

**Figure 12. Water balance of Lake Titicaca for the period 1982–2016 in terms of (a) interannual variability and (b) seasonal cycle. The values in parentheses correspond to the mean annual or monthly values (in mm) for the period 1982–2016. The lake water balance follows the equation** $P_{lake} + Q_{in} - E_{lake} - Q_{out} - dh/dt + \varepsilon = 0$

## 5 Discussion and conclusions

### 5.1 Main findings

In this study, the hydrologic processes that occur in a large lake hydrosystem were simulated using an integrated modeling framework. The aim of the simulation was find a balance between serious problems such as data availability, the complexity of representing the processes, and reliability of predictions despite uncertain and incomplete observed data. The originality of this modeling framework lies in its ability to integrate and simulate natural hydrologic processes and irrigation in a poorly-gauged large-lake hydrosystem at a daily time step. Unlike other studies (e.g. Hosseini-Moghari et al., 2020) that attempted to integrate natural hydrological processes and anthropogenic pressures in large lakes, our study takes a long-term approach making it possible to understand variations in water balance components over a multidecadal period. This enables a better understanding of the drivers of variations in water level under a wide range of climate variability conditions and anthropogenic pressure. In addition, our modeling framework includes the majority of components of the water balance (direct precipitation over the lake, evaporation from the lake, upstream inflow, and downstream outflow) except net groundwater exchanges. While Hosseini-Moghari et al. (2020) focused on estimating upstream inflow to Lake Urmia, they paid little attention to other components such as direct precipitation over the lake and evaporation that have a major impact on the water balance of large



lakes, as shown in the current study. Our results for Lake Titicaca show that the modeling framework makes it possible to capture variations in the lake water level reasonably well. Our simulations are significantly more accurate than those of Lima-Quispe et al. (2021), who overestimated water levels during the predominantly dry period of the 1990s. In contrast to the previous study, our modeling framework was implemented at a daily time step and included high mountain hydrological

processes such as the contribution of snow and ice, as well as a better representation of irrigation thanks to new data on irrigable area. In addition, our estimates of reference evapotranspiration and evaporation from the lake capture the non-stationarity of the climate, whereas the Lima-Quispe et al. (2021) study used monthly average data for some climatic variables (relative humidity, solar radiation and wind speed). These improvements resulted in more accurate water level simulations that adequately reflect long-term variability in the face of a changing climate. The main contributions of this study are detailed

below.

Our approach to estimate upstream inflow considers the natural hydrological processes (including snow and ice accumulation and melt) and irrigation including water allocation. One advantage of this type of approach is that water requirement and allocation in the face of climate variability are implicitly simulated, which is not the case when fixed water requirement values are imposed. The approach we used improved our understanding  of the role of tropical glaciers and irrigation in the

hydrological functioning of Lake Titicaca. Hydrologic sensitivity analysis revealed that snow and ice processes had minimal an impact on the predicted streamflow in the catchments that contribute to the lake. Also, the water balance presented in Table 5 shows that the proportion of snow and ice melt is very small compared to the proportion of rainfall at the scale of the whole Titicaca hydrosystem. This suggests that the study area is dominated by a rainfall regime. Therefore, the role of snow and ice melt in the functioning of the lake could be considered negligible. That said, we do not intend to underestimate the contribution

of glaciers. In fact, they are relevant at the scale of the headwater catchments, especially for the supply of water to large cities such as El Alto and La Paz (Buytaert et al., 2017; Soruco et al., 2015), maintenance of wetlands like the *bofedales* (Herrera et al., 2015), and for irrigation (Buytaert et al., 2017). The impact of net irrigation consumption was also shown to be of little significance for streamflow prediction. In most of the gauged catchments, only minimal improvements in the modeling performance were achieved when irrigation was taken into account. This approach nevertheless made it possible to estimate

the net consumption due to irrigation at the scale of the catchments that contribute to the lake. Even if this consumption remains low compared to consumption of water used for other purposes, it is likely to increase significantly in the future linked to the climatic and anthropogenic changes the study area may undergo. It should also be noted that this process was not accounted for solely on the basis of the soil water deficit and that local knowledge of farmers' practices in terms of irrigation scheduling was also taken into account. Although some authors (e.g. Githui et al., 2016) mentioned that such information is readily

available, this is not entirely true in contexts such as the Titicaca hydrosystem, where traditionally managed small irrigation systems predominate. However, incorporating the details of water management of the numerous irrigation systems is not feasible at the scale of the upstream catchment of large lakes.

The periods of rising and falling water levels are closely linked to direct precipitation over the lake and upstream inflow (see Fig. 12a), which is mostly influenced by interannual precipitation variability. As such, understanding the effects of climate



oscillations on precipitation variability is crucial to understand water level. Some authors (e.g. Garreaud and Aceituno, 2001; Jonaitis et al., 2021) noted that the interannual variability of precipitation in the region is mainly driven by the El Niño Southern Oscillation (ENSO). During its warm phase conditions are usually dry, while during the cold phase conditions are usually wet, although this relationship is not always absolutely true (Garreaud et al., 2003). For example, Jonaitis et al. (2021) observed negative precipitation anomalies during the La Niña phase and positive anomalies during the El Niño phase in the Lake Titicaca region, although they note that the anomalies were not statistically significant. Segura et al. (2016) argue that El Niño plays an important role in the interannual precipitation variability and that decadal and interdecadal variations are influenced by sea surface temperature (SST) anomalies in the central-western Pacific. Consequently, variations in water level cannot be attributed to ENSO alone, but may also be influenced by other climatic oscillations. Sulca et al. (2024) found that interannual variation in water levels is related to SST anomalies in the southern South Atlantic, and that this interdecadal and multidecadal variability can be explained by Pacific and Atlantic SST anomalies. Furthermore, they noted that multidecadal variation is related to North Atlantic SST anomalies and the southern South Atlantic SST anomalies. Complex teleconnections between multiple climate oscillations and variation in the components of the water balance (including water levels) were also observed by Saber et al. (2023) in the Great Lakes of North America. These authors found that the effect of climatic oscillations on the water balance of lakes mostly occurred at interannual to interdecadal scale. It would be interesting to analyze the influence of atmospheric teleconnections on water levels in Lake Titicaca, like in the study of Saber et al. (2023) but it was clearly beyond the scope of the present study.

Annual evaporation (1,616 mm yr$^{-1}$), the biggest component of the lake water balance, is in the same range as the evaporation (~1,600) of other lakes located at low latitudes (Wang et al., 2018) and also within the ranges reported in previous studies of Lake Titicaca (~1,600) (Delclaux et al., 2007; Pillco Zolá et al., 2019; Lima-Quispe et al., 2021). The evaporation rate is quite high (1,616 mm yr$^{-1}$), even though the air temperature over Lake Titicaca is low due to its high altitude. Thus, evaporation depends mostly on net radiation, although humidity, wind speed, and changes in heat storage play an important role in the seasonal variation. Regarding heat storage change, the maximum heat gain of the lake is reached in October and the maximum heat loss in May (Fig. 13a). Neglecting changes in heat storage results in overestimation of evaporation in the months the lake is heating and in underestimation of evaporation during the months the lake is cooling (Fig. 13b). The same result was also reported by Bai and Wang (2023) for Lake Taihu in China. Although several studies on evaporation from Lake Titicaca have been carried out using a variety of methods (Carmouze, 1992; Delclaux et al., 2007; Pillco Zolá et al., 2019), our estimates are innovative because they are long term but also cover a recent period. Moreover, the accuracy of our estimates was evaluated in the context of the water balance, which adds an additional level of reliability to our findings.



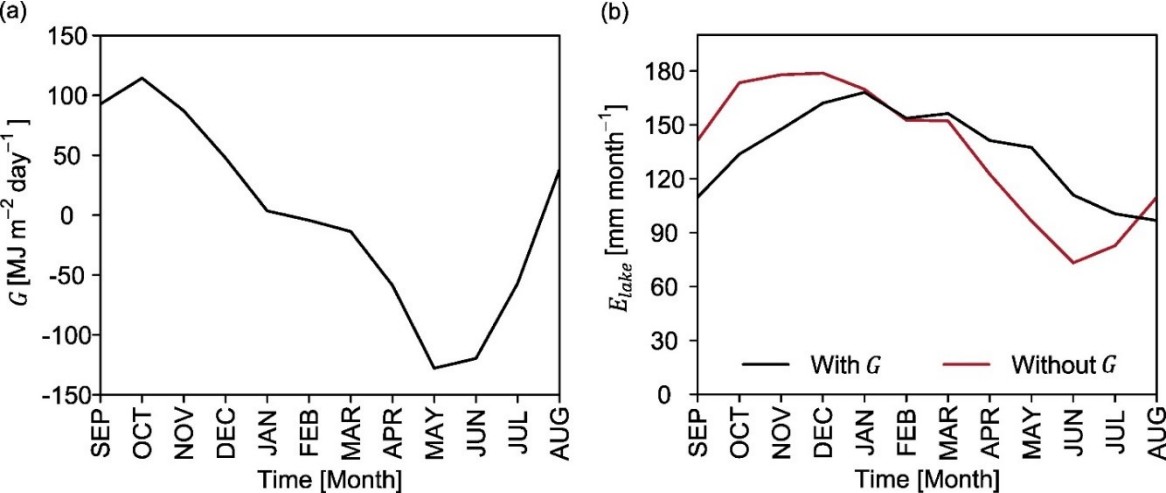

**Figure 13. Seasonal variations in (a) heat storage and (b) the role of heat storage in seasonal variation of evaporation.**

**5.2 Limitations of the modeling framework**

The results provided by the proposed modeling framework may be affected by a number of uncertainties and limitations that are discussed below.

Model forcing and evaluation data are the main sources of uncertainty. Daily precipitation and air temperature from GMET

rely strongly on the spatial representativeness of ground-based measurements that are very sparse at elevations above 4,000 m a.s.l. As a result, the estimated snow and ice processes above that elevation may be affected by significant inaccuracies. In part, this could explain the poorly simulated mass balances at some glaciers compared to geodetic observations (which are also uncertain). As shown by Ruelland (2020), the lack of high-altitude stations can have serious implications for correct estimations of precipitation and temperature lapse rates, and thus the difficulty of realistically representing the accumulation and ablation

processes. It is difficult to assess the reliability of GMET based on hydrological sensitivity analyses in high mountain areas because measured streamflow is not available at the headwater scale. The precipitation estimated by GMET on the lake can also be questioned because it is only based on a few stations located on certain islands, which does not guarantee they correctly represent local convective phenomena (Gu et al., 2016; Nicholson, 2023) due to the very large surface area of the lake. Consequently, the underestimation of lake water levels in the early 2000s may be associated with underestimation of

precipitation.

The simulation of snow and ice processes also needs discussing. The degree-day approach used in the present study might not be perfectly suited to simulate the melting of snow and ice at daily time steps in tropical glaciers, where melting energy is not always correlated with air temperature (Sicart et al., 2008; Rabatel et al., 2013). It may be more appropriate to use models based on energy balance, although they require large quantities of data. For instance, Frans et al. (2015) applied the DHSVM

model driven by reanalysis data fitted with local measurements in the Zongo catchment. For streamflow prediction over the period 1992–2010, while the performance they obtained at the monthly time step was satisfactory, this was not the case when





the evaluation was conducted at a daily time step. The authors argue that the poor performance was due to inaccuracies in the forcing data. This suggests that using a more sophisticated model does not necessarily lead to more realistic streamflow simulations and that uncertainty in the input data may be more important than the structure of the model used. Another

limitation is the direct transfer of snow and ice model parameters obtained in the Zongo catchment to the whole Titicaca hydrosystem. Although temperature index-based model parameter transfer has been shown to work relatively well, i.e., with only a small reduction in model performance in southern Swiss Alps (Carenzo et al., 2009) and in western Canada (Shea et al., 2009). The unsatisfactory performance of mass balance simulations on some glaciers may rather be due to inaccurate precipitation and air temperature data. On the other hand, the estimated mass balances did not consider changes in glacier area,

thickness and volume over time. This is a significant limitation in terms of capturing non-stationary conditions. As the surface area of the glaciers in our model was obtained in 2000 and considering glacier shrinkage worldwide, melting before the year 2000 may be underestimated, and after the year 2000, it may be overestimated. However, on the Zongo glacier, no significant differences were observed between the simulated and observed mass balance. In any case, it is worth noting that the assumptions made to simulate the glaciological functioning actually have minimal influence on the water balance of Lake

Titicaca.

Estimates of lake evaporation and reference evapotranspiration have also certain limitations that are worth mentioning. Reanalysis data were used for some forcing data (humidity, solar radiation and wind speed). Wind speed data were adjusted using the bias found at one meteorological station, which is not necessarily representative of the entire spatial domain. However, we believe that the impact is not very significant because the aerodynamic component accounts for only about 20%

of the reference evapotranspiration and evaporation. This is consistent with previous research in the study area, which indicated that, rather than the aerodynamic component, it is the radiative component that contributes significantly to reference evapotranspiration (Garcia et al., 2004) and evaporation from the lake (Delclaux et al., 2007). The predictive uncertainty of actual evapotranspiration is small, suggesting that the catchment streamflow predictions are not very sensitive to the reference evapotranspiration. Concerning evaporation, the estimated change in heat storage is based on certain assumptions that could

be questionable. Instead of water temperature at different depths, we used lake surface water temperature (LSWT), as suggested by Pillco Zolá et al. (2019), and the magnitude and seasonal variation in our evaporation estimates are in agreement with those of Carmouze (1992) based on temperature measurements taken at different depths.

Concerning irrigation, the main limitations are linked to the stationarity of the irrigable area and crop types. Better model performance (albeit marginal) was obtained when irrigation was included. However, we did not consider changes in irrigable

area. Some authors (e.g. Geerts et al., 2006) reported an increase in quinoa production in the Altiplano. However, quinoa is usually a rainfed crop and consequently requires limited irrigation. Information was also missing concerning changes in the crops cultivated over time. $Kc$ derived from satellite images (Pôças et al., 2020) could be an interesting way to fill this gap. However, small irrigation systems predominate in our study area, which could have implications for the accuracy of the estimates. In addition, the location of the inventoried irrigation systems is referential, since there are no maps showing

delimited areas and it would consequently first be necessary to delimit the irrigable area and then to use spectral indices to



derive the crop coefficients. Despite these limitations, it should be noted that reference evapotranspiration was driven by time-varying meteorological data. The water requirement is thus not entirely stationary.

Although WEAP has the ability to dynamically simulate groundwater-surface water interactions (e.g. Dehghanipour et al., 2019), they were not physically modeled due to lack of forcing and control data. The contributions of groundwater in the

catchments were only simulated with the deep store of the SMM. This is an important simplification because it neglects the dynamic interaction between rivers and aquifers. The use of pumped groundwater for irrigation was also not taken into consideration because irrigation inventories (Ministerio de Medio Ambiente y Agua, 2012) indicated that the proportion of groundwater in the supply was very small. Therefore, we would expect the impacts on streamflow prediction and water supply results to be negligible. Groundwater-lake interaction was also neglected. The simulated and observed lake water levels showed

small discrepancies, suggesting that the net groundwater exchanges are slight compared to interactions among other components of the water balance.

## 5.3 Prospects

Despite the aforementioned limitations, our modeling framework and its associated results on the water balance components will be useful to support decision making in water resources management in Lake Titicaca because they represent climatic,

glacio-snow-hydrological, and water allocation components. Up to this stage, the approach has helped us to quantify the hydrological cycle and thus understand the hydrological functioning of Lake Titicaca. In the next stage, water management scenarios could be evaluated in the context of climate change. Some of the management scenarios that could be explored include increasing the irrigable area, the efficiency of irrigation systems, and releases from the lake. However, it is first necessary to remedy certain existing shortcomings in the modeling framework, such as changes in the surface area and volume

of glaciers, drawing inspiration from simple approaches that can be found in the literature (e.g. Seibert et al., 2018). To initialize the model, global glacier thickness datasets could be used (e.g. Farinotti et al., 2019; Millan et al., 2022). On the other hand, the design of water management scenarios would require collaborative work with stakeholders, in particular with the Lake Titicaca Authority. Exploring future scenarios will be crucial to identify and plan intervention actions to ensure the sustainability of Lake Titicaca. Such experiments could also serve as a replicable model for other large lakes around the world.




## Appendices

### Appendix A. Performance of the ERA5-Land dataset

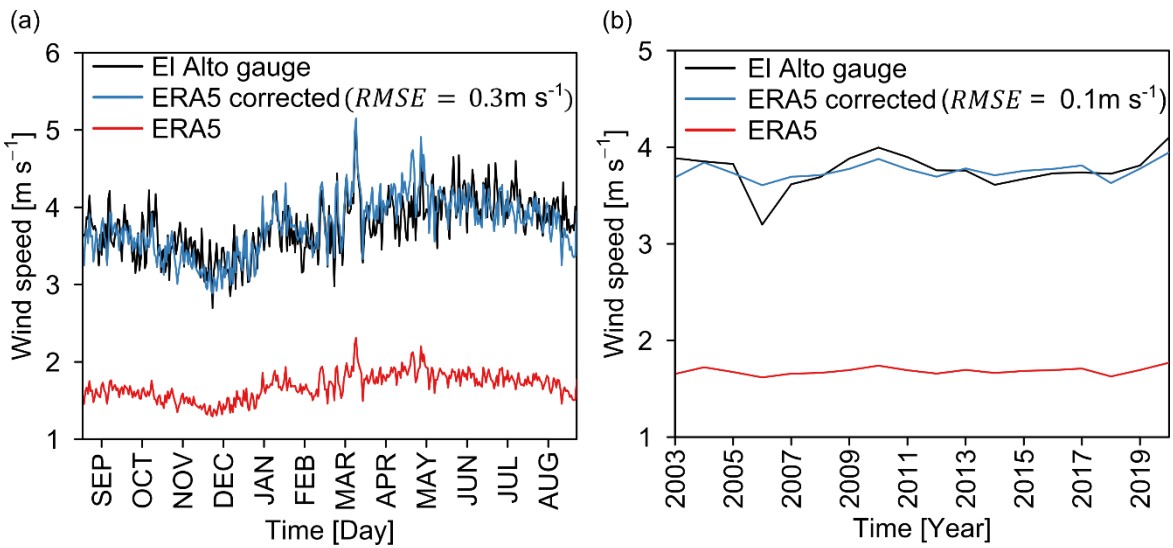

**Figure A1. Wind speed performance of ERA5 evaluated at El Alto gauge in terms of (a) seasonal cycle and (b) interannual variation for the period 2003-2020.**

### Appendix B. Simulation of lake water surface temperature (LSWT)

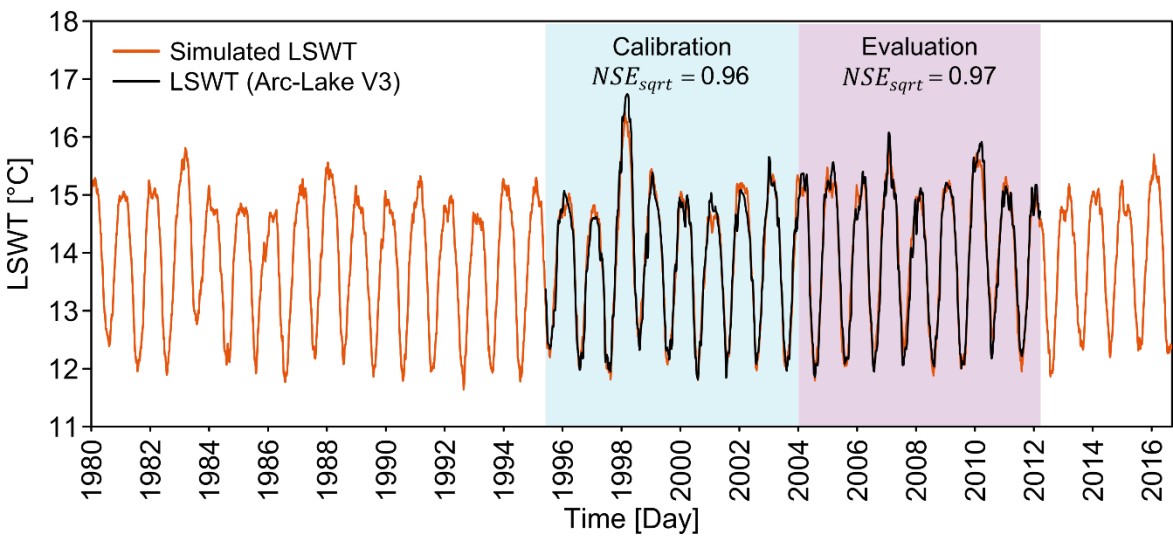

**Figure B1. Lake surface water temperature (LSWT) simulated with the Air2wateR model (Piccolroaz et al., 2013; Toffolon et al., 2014) for Lake Titicaca and its calibration and evaluation performances.**





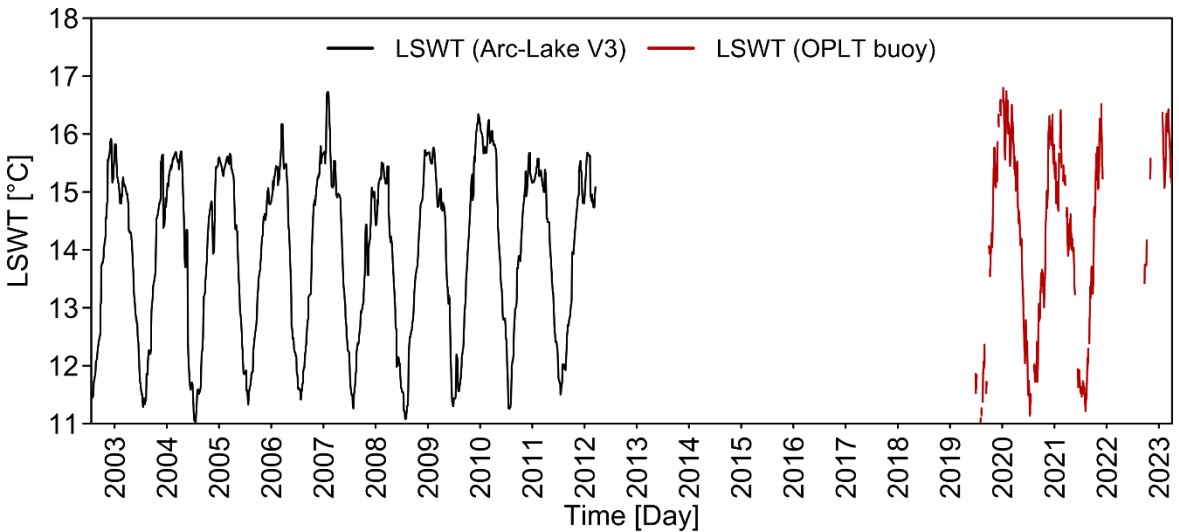


**Figure B2. Lake surface water temperature (LSWT) obtained from the buoy site (16.25ºS, 68.68ºW) in Arc-Lake V3 (MacCallum and Merchant, 2012) and OPLT (Lazzaro et al., 2021)**

**Appendix C. Simulated and observed water levels**

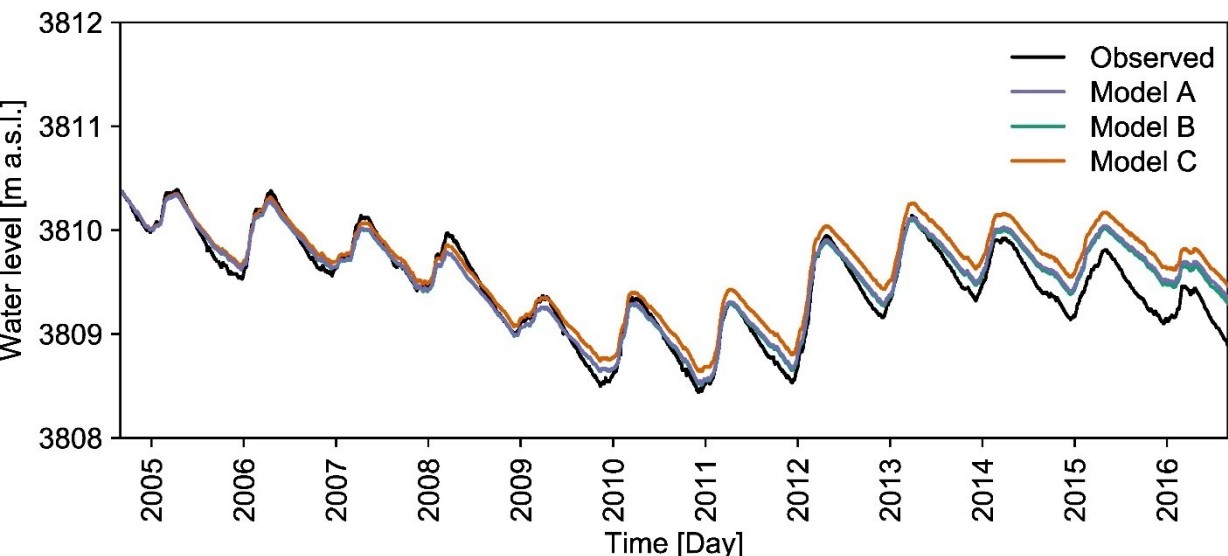

**Figure C1. Performance of the modeling chain compared with the water levels of Lake Titicaca measured in Puno. Three simulations of water levels are presented because the models produced three different estimates of $Q_{in}$.**




*Data availability.* The data used in this study are available at their respective websites: GMET (can be accessed directly at WEAP), ERA5-Land (https://cds.climate.copernicus.eu/cdsapp#!/dataset/reanalysis-era5-land), CLACIOCLIM (https://glacioclim.osug.fr/Glacier-du-Zongo-127), Observatorio Permanente del Lago Titicaca (https://olt.geovisorumsa.com/Datos.html), geodetic mass balance (https://www.theia-land.fr/en/product/rate-of-glacier-

elevation-changes-from-2000-to-2019/) and ARC-Lake v3 (https://researchdata.reading.ac.uk/186/). Measured streamflow and lake water levels should be requested from SENAMHI-Peru and SENAMHI-Bolivia.

*Author contributions.* NLQ, DR and TC conceptualized and designed the study approach. AR provided support in the methodological design of the glacier simulation. WLC provided the hydrological and climatic data for the Peruvian side. NLQ

and DR implemented the integrated model and analyzed the simulations, and prepared the original version of the manuscript. All the authors contributed to reviewing and editing the manuscript.

*Competing interests.* The authors declare they have no conflict of interest.

*Financial support.* This research was funded by AFD and IRD under the CECC project (Water cycle and climate change) in the framework of a PhD funding. This study is also conducted as part of the International Research Network (IRN) ANDES-C2H, a joint initiative of the IRD and universities and institutions in Colombia, Bolivia, Peru, Ecuador, Chile, and Argentina.

*Acknowledgements.* The authors are grateful to IRD for funding the study. Our special thanks to SENAMHI-Peru and

SENAMHI-Bolivia for providing the ground-based hydroclimatic data. We also thank Javier Núñez (IIGEO-UMSA) and Xavier Lazzaro (IRD) for providing the data from the buoy of the Permanent Observatory of Lake Titicaca. Finally, we thank other data providers including GLACIOCLIM (for Zongo glacier data), THEIA (for access to geodetic mass balance data), ESA (for access to ARC-Lake V3 data), ECMWF (for access to ERA5-Land data), and SEI (for access to GMET data).





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
