# Peer review of "Assessing the different components of the water balance of Lake Titicaca"

_EGUsphere, 2024_

## Author Comment (AC1)

**Responses to comments from anonymous Reviewer #1**

On "Assessing the different components of the water balance of Lake Titicaca" by Nilo Lima-Quispe
Denis Ruelland, Antoine Rabatel, Waldo Lavado-Casimiro, and Thomas Condom.

**General comments**

The manuscript presents a water balance modeling chain for Lake Titicaca, simulating various water balance components, including precipitation, evaporation, inflow (considering irrigation water abstraction and glacial contributions), and outflow. The model effectively closes the water balance and quantifies the contributions of these components. Notably, the authors find that glaciers and irrigation water abstraction have a minimal impact on the overall results.

We would like to sincerely thank the referee for the time and effort she/he spent reading the initial manuscript and for making many clear, pertinent and constructive suggestions for improvement. This greatly helped to highlight the novelty and ensure clarity of the paper. Comments are shown in black text, while our responses appear in blue, with suggested modifications emphasized in *blue italics*.

The manuscript is interesting, well-written, detailed, and comprehensive and of interest for publication in HESS. However, the text is at times overly lengthy and repetitive, particularly in the methods and results sections. Condensing the content without sacrificing critical information and possibly moving some material to supplementary sections would enhance clarity and focus. In addition, the level to which the presented manuscript is build upon and novel to Lima-Quispe et al., 2021 in terms of methods applied can be made a bit more explicit. To aid the authors with this, detailed comments and suggestions are provided below.

We have carefully addressed the comments. Our responses can be found in the sections for general and specific comments.

**Major Comments**

- The manuscript concludes that glaciers and irrigation have a minor impact on the water balance. Given this finding, the detailed analysis of these components could be relocated to the appendix to streamline the manuscript.

  A specific objective is to evaluate the hydrologic sensitivity of upstream inflow predictions (including lake water level variations) to net irrigation withdrawals as well as snow and ice processes. The findings indicate that these factors have minimal impact. Nevertheless, this could be demonstrated precisely because these factors were considered in the modeling chain. However, we acknowledge that certain details related to data, methods, and results may affect the readability of the article. To address this, we propose the following changes:

  - **Materials:** Sentences referencing the Zongo glacier have been moved to the appendix. We have also shortened the sections on agricultural characteristics and irrigation management.
  - **Methods:** The procedure for obtaining snow and ice model parameters using the Zongo catchment has been moved to the appendix.
  - **Results:** We have shortened the paragraphs discussing the performance of the snow and glacier model compared to the geodetic mass balance.

- The manuscript builds on the model presented by Lima-Quispe et al., 2021. For example, the climatological data used (from the GMET tool) are very similar. This should be more clearly

described in the manuscript, taking Lima-Quispe et al., 2021 as a starting point and not presenting already published methods as novel. The authors should more explicitly differentiate their work from the previous study, highlighting the findings of the former, and showcasing how the current study builds upon it. The novelty is already highlighted to some extent, for example by articulating the contribution of a detailed glacier method. However, the direct contributions from this have very little impact on the final results. Seen both studies are authored by the same first author, a direct comparison between the results of both studies, possibly through a figure, would further clarify the advancements made in the current study (for example, figure 6 of Lima-Quispe et al., 2021 is very similar to Fig 10 of the current paper).

We acknowledge that the current study utilizes some models from the WEAP platform previously presented by Lima-Quispe et al. (2021) at a monthly time step. There is also overlap in the precipitation and air temperature data (from the GMET tool), as well as in the irrigable area data on the Bolivian side. In terms of methods, there is overlap in the rainfall-runoff model (Soil Moisture Model), but in the current study, we simulate at a daily time step and employ a different approach to simulate irrigation water allocation as well as a snow and ice module which were not used in the first study, making it impossible to estimate the contribution from snow melt and ice melt to lake water level's variations. Additionally, to apply the model at the daily time step, it was necessary to collect new data updated to the required time scale. Furthermore, new data, such as lake bathymetry and irrigable area on the Peruvian side, became available.

We also acknowledge that, in the submitted version of the manuscript, we have not sufficiently highlighted the novelty of our study compared to the previous work by Lima-Quispe et al. (2021). A significant limitation of the previous study is that the authors applied a scaling factor to precipitation, in combination with calibrating the constant of a power function ($f(x) = ax^b$) to estimate the rating curve for lake outflows in an attempt to close the lake's water balance. For instance, Equation 1 in Lima-Quispe et al. (2021) does not account for the error or residual term in the water balance. We believe that applying scaling factors introduces additional uncertainty. A reliable water balance should close with a very small error or residual, without the need for scaling factors (Rientjes et al., 2011; Vanderkelen et al., 2018). Other limitations of the Lima-Quispe et al. (2021) study that we can highlight are:

1. The omission of snow and ice processes, which may play a crucial role in this high mountain region. To date, no study has assessed the impact of snow and ice melt on lake water level fluctuations. For instance, during periods of water level decline, some stakeholders often attribute this to glacier retreat and irrigation water withdrawals. Therefore, assessing the hydrologic sensitivity to these processes is essential.
2. Estimating evaporation using the Penman method, without accounting for changes in heat storage. Additionally, the authors estimated the lake surface water temperature (LSWT) by simply adding a delta value to the air temperature. Given that Lake Titicaca is located at an average altitude of 3810 m a.s.l., radiation plays a crucial role in the lake's physical and chemical processes. Therefore, it is essential to consider both LSWT and changes in heat storage. Since evaporation is the largest component of the water balance, special attention must be given to estimating it as accurately as possible, using available data, whether through in situ measurements, remote sensing, or reanalysis.
3. The use of historical monthly averages (humidity, wind speed, and incoming solar radiation) to estimate evapotranspiration and reference evaporation, without accounting for interannual variability. In light of ongoing climate change, it is crucial to consider temporal

variations in these variables. This could have significant implications for evaporation estimates and, consequently, the accuracy of the water balance. While data in the region are scarce, reanalysis data can now be used to improve these estimates.

One of the main novelties of our study is that the modeling approach allows us to close the water balance of Lake Titicaca with a very small error, without the need to apply scaling factors. The daily water levels simulated over the period 1982–2016 show acceptable performance. In the following figure, we present a comparison (at a monthly time step) of the water level simulations from our study and those of Lima-Quispe et al. (2021), without applying a scaling factor to the precipitation over the lake. As shown, the simulations by Lima-Quispe et al. (2021) without the scaling factor result in unrealistic water levels, mainly after 2000. For this reason, the authors increased the precipitation in an attempt to close the water balance.

[Figure]

Figure 12. Comparison of simulated water levels by BasicModel+IRR+SNOW/ICE (aggregated to monthly time step) with those obtained by Lima-Quispe et al. (2021) without applying the scaling factor to the precipitation over the lake.

Our modeling framework benefits from the following: (i) a rigorous calibration and evaluation procedure to simulate upstream inflow (ii) the estimation of lake evaporation using the Penman method, which accounts for LSWT (estimated using a combination of a conceptual model and remotely sensed data) and heat storage; and (iii) estimates of reference evapotranspiration and lake evaporation that capture climate variability using the ERA5-land dataset for humidity, wind speed and solar radiation.

Another novelty is that we demonstrate that net irrigation withdrawals and snow and ice melt have minimal influence on water level variations. We also disentangle the role of changes in heat storage in estimating evaporation.

For the reasons stated above, our modeling approach is innovative, though it does share some overlaps in specific models (soil moisture model) and input data (GMET and irrigable area in the Bolivian side).

To address the reviewer's comment, we propose the following changes:

- In the Introduction section, before the objectives, we added a section introducing Lake Titicaca, where we mention the previous study and its limitations in order to better contextualize and highlight the novelty of our study.
- In the Methods section, we acknowledge the overlaps in methods and data with the previous study (Lima-Quispe et al. 2021).
- In the Discussion section, we clearly present the novel aspects of our study in comparison to the previous one.

The changes in the text are reflected in our responses to the specific comments.

- The manuscript would benefit from a more concise presentation. Repetitions, and lengthy wordings in the introduction, methods and results should be reduced.

Agreed.

We acknowledge that the initially submitted version was sometimes repetitive and lengthy. We have identified and removed the repetitions and shortened certain paragraphs to make it more concise. We propose the following changes:

- **Introduction:** This section was originally about 2,400 words, now reduced to 1,900 words. Section 1.2 (Methods to estimate the individual components of a lake water balance) was too long and has been shortened. We also removed Section 1.3 (Integrated water balance in large lakes) and instead introduced Lake Titicaca by explaining why this study area was chosen and addressing the limitations of previous studies.
- **Materials:** The original version had about 2,700 words, now reduced to 1,700 words. Some paragraphs related to the context of Lake Titicaca were moved to the Introduction section. We also shortened the sections related to irrigation. Details on the evaluation of the ERA5-Land and Zongo Glacier data were moved to the Appendix.
- **Methods:** Initially 4,300 words, now reduced to 3,500 words. Sections related to the modeling of the Zongo catchment and LSWT have been moved to the Appendix.
- **Results:** Originally 3,000 words, now reduced to 2,600 words. We shortened the glacier model performance section and removed repetitions.
- **Discussions:** Originally 2,800 words, now reduced to 2,200 words. We removed sentences that were already mentioned earlier in the text.

- Abstract and Introduction: The introduction currently provides a broad overview of large, poorly gauged lakes, but since the focus is on Lake Titicaca, it should be emphasized from the outset. The introduction would benefit from a revision to prioritize the unique aspects of Lake Titicaca, with general information on lakes serving as background rather than the main focus. The description in the introduction of lakes in general terms could be condensed, with more specific examples of how different lakes' conditions can vary. This will help contextualize the study's focus on Lake Titicaca.

We agree it is appropriate to introduce Lake Titicaca earlier in the Introduction section. We have thus reorganized the Introduction according to the following sections:

- **1.1 On the need for an integrated water balance in large lakes:** We discuss variations in lake water levels and emphasize the need to estimate water balances that account for both natural flows and water management to better understand the drivers of these variations. We then introduce the concept of integrated modeling and highlight how it differs from traditional approaches.

- **1.2 Challenges in estimating the water balance of large lakes:** We outline the challenges involved in estimating the individual components of the water balance, which is essential for placing our research within a broader context.
- **1.3 Placing Lake Titicaca in the context of an integrated water balance:** We introduce Lake Titicaca, focusing on its unique characteristics and the challenges it faces. We also describe the lake's fluctuations and the need for a reliable water balance to support decision-making. Additionally, we present the previous study by Lima-Quispe et al. (2021) and highlight its limitations.
- **1.4 Scope and objectives:** We present the research questions and objectives.

We hope these changes address the reviewer's comment.

**Specific comments:**

L42: Specify which lakes are being referred to as "many lakes."

Agreed. The sentence has been modified to: *"…For example, it has been observed that direct precipitation over the African Great Lakes is more intense than in their surrounding areas…"*

L81: Replace "water uses" with "water management" to include aspects like dam management, which could lead to differences in flow magnitude and seasonality.

Agreed. The change was made.

L84-86: Clarify and reformulate the example given.

This sentence was removed in order to reduce the Introduction and focus more on integrated modeling and Lake Titicaca.

L124: Clarify whether "net water withdrawals" refer to the lake itself.

Agreed. In this case, it refers to the contributing catchments. The sentence has been modified to: *"…Several studies on large lakes (>500 km2) (e.g., Rientjes et al., 2011; Vanderkelen et al., 2018; Wale et al., 2009) have estimated the water balance under the assumption that net water withdrawals in the contributing catchments are negligible…"*

Section 1.4: Explain why Lake Titicaca was chosen for this study and what value the water balance study brings to lake managers.

Agreed. To address this comment, we have reorganized and condensed the content of the Introduction. We created a new Section 1.3 (before the objectives and scope) to introduce Lake Titicaca. In this section, we justify the selection of the study area by highlighting that it is a poorly gauged, transboundary lake with complex water management challenges that require a reliable water balance estimate. Information previously presented in the study area section was used to write this paragraph.

The proposed paragraph is presented here: *"Lake Titicaca, located on the Altiplano of the tropical Andes of South America, is one of the highest large lakes in the world and an interesting case study for an integrated water balance. This lake is part of a vast endoreic catchment, and is connected by the Desaguadero River to Lake Poopo in Bolivia (Lima-Quispe et al., 2021). As a transboundary lake shared by Peru and Bolivia and a poorly-gauged hydrosystem, it faces many of the aforementioned challenges. The region experiences significant interannual climate variability (Garreaud and Aceituno, 2001), which, coupled with complex water management issues (Revollo, 2001), intensifies the difficulties of managing Lake Titicaca. These*

*challenges include extreme hydrological events (droughts and floods), lake releases, and water pollution (Revollo, 2001; Rieckermann et al., 2006). Water levels measured in Puno (Peru) have fluctuated by approximately six meters over the past century, with the lowest recorded in 1943–1944 and the highest in 1984–1986 (Sulca et al., 2024), causing US$125 million in flood damage (Revollo, 2001). In response to these challenges, a management plan was developed in the early 1990s for both Lake Titicaca and the Altiplano hydrosystem (Revollo, 2001). A key component of this plan was the construction of an outflow gate to regulate lake releases and the establishment of operating rules. Although the outflow gate was completed in 2001, lake releases remain nearly the same as under natural conditions because the operating rules have not yet been implemented. Addressing these water management challenges requires an accurate integrated water balance allowing a better knowledge on the drivers of the lake water level variations."*

L155: Introduce Lima-Quispe et al., 2021 earlier in the introduction, detailing its findings and limitations.

Agreed. To address this comment, we have reorganized the contents of the Introduction. This involved reducing the length of several paragraphs. Specifically, we have added a new paragraph before the objectives and scope, where we discuss the limitations of the study by Lima-Quispe et al. (2021). This revision helps better position our study in relation to previous research.

Here is the new paragraph: *"Unlike other large lakes, very few studies have been conducted on Lake Titicaca. The only study that modeled the water balance of Lakes Titicaca and Poopó was the one by Lima-Quispe et al. (2021) using the Water Evaluation and Planning System (WEAP) platform with a monthly time step for the period 1980–2015. The study aimed to distinguish the relative contributions of climate and irrigation management on water level fluctuations. However, the modeling approach proposed by the authors has a significant limitation because it is based on a scaling factor for precipitation over the lake to close the water balance, which clearly introduces additional uncertainty. Other methodological shortcomings include: (i) the omission of snow and ice processes, which can play a non-negligible role in this high-elevation region; (ii) the estimation of evaporation using the Penman method, without accounting for changes in heat storage; and (iii) the use of historical monthly averages (humidity, wind speed and incoming solar radiation) to calculate reference evapotranspiration and evaporation, without considering interannual variability."*

L161: why is this an "integrated" water balance approach and what is the difference with a normal water balance approach?

To address this comment, we have added a paragraph defining integrated water balance modeling:

*"In large lakes, it is essential to adopt integrated water balance modeling, which represents the interactions and feedbacks between natural hydrological processes and water management within a single modeling framework (Niswonger et al., 2014). Unlike traditional decision support systems applied to large lakes (Hassanzadeh et al., 2012), which typically simulate natural flows and irrigation water requirements independently, integrated modeling enables these processes to be simulated in a coupled and dynamic manner."*

L169: Include the lake's surface area.

Agreed. Although the lake area was mentioned at the beginning of the study, we believe it was not entirely clear. To clarify, we have revised it to: *"Lake Titicaca, located at 3,812 m a.s.l. on the Altiplano of the tropical Andes of South America, covers an area of approximately 8,340 km² and spans the borders of Peru and Bolivia."*

Section 2.2: Acknowledge the methodological overlap with Lima-Quispe et al., 2021.

Agreed. We did use the same set of precipitation and air temperature data. We have clarified this in the text with the following modification:

*"...It is based on a probabilistic method using ground station data (for more details see Clark and Slater (2006), and Newman et al. (2015)). Lima-Quispe et al. (2021) used the same data."*

Figure 2, panel b: Adjust the color bar to ensure white corresponds to zero or use a sequential color bar.

Agreed. To address the reviewer's comment, the figure has been modified using diverging colors, with yellow corresponding to zero Celsius degree. Here is the revised figure:

[Figure]

L252: Explain "Crop coefficient Kc" when first introduced; consider moving this to the methods section.

Agreed. We have moved the sentence related to "Crop coefficient Kc" to the Methods section, where the irrigation modeling approach is presented. The revised text is as follows:

*"In SMM, water requirements (WR) for irrigation are determined by crop evapotranspiration (calculated from seasonal crop coefficients (Kc) and reference evapotranspiration) and the depletion of available water in the root zone store (see Fig. 4). Kc adjusts the reference evapotranspiration (ET0) to reflect crop-specific characteristics, such as phenology (Allen et al., 1998). It was derived using cropping calendar and crop type data (Autoridad Nacional del Agua, 2009, 2010; Instituto Nacional de Estadística, 2015)."*

L347: Clarify what is meant by "production."

Agreed. Production (as well as routing) is a commonly used term in conceptual hydrological modelling. However, to address the reviewer's comment and make the purpose clearer, the sentence has been modified as follows: *"...the processes contributing to the generation and regulation of water storage and water flow in the catchments…"*

L445: Specify if LSWT time series availability is meant here.

Agreed. We have modified the text to include the following sentences: *"Since there are no long-term measurements of lake surface water temperature (LSWT) and the remotely sensed data sets do not cover the entire study period, the Air2Water model (Piccolroaz et al., 2013; Toffolon et al., 2014) was used to simulate LSWT."*

Additionally, details on the LSWT simulation have been moved to the appendix to improve conciseness.

L475: Replace "assessment" with "evaluation."

Agreed. The word was changed.

Models A, B, C: Consider renaming these to something more descriptive (e.g., IRR+GLAC) to avoid repetitive explanations and ease the figure interpretations.

Agreed. We have renamed the three modeling options as follows: *"…Three model structures were tested under the following configuration: (i) BasicModel+IRR+SNOW/ICE, which represents the full reference model structure (as shown in Fig. 4); (ii) BasicModel+IRR, where the processes associated with snow and ice (accumulation and ablation) are excluded, but irrigation is maintained; and (iii) BasicModel, where both snow and ice processes and irrigation consumption are excluded…"*

Changes reflecting these revisions have been made throughout the text, as well as in the figures and tables.

L568-573: Rewrite these sentences to be more concise.

Agreed. We have refined the writing to make it more concise. The modified sentences are:

*"… The scatter plots reveal significant variability in model performance, with some glaciers (represented by each point) close to the identity line and others deviating significantly. The model simulates a more negative mass balance compared to the geodetic mass balance. Figure 7a displays glaciers according to their surface area, while Figure 7b shows them based on their mean elevation…".*

Note that the numbering of the figures has been adjusted because some were moved to the appendix.

L595: Indicate in the figure caption that it deals with "glacier" mass balance; same for Table 4.

Agreed. Both the figure caption and the table have been clarified regarding glacier mass balance. For example:

*"Figure 7. Scatter plots comparing simulated and geodetic glacier mass balance for 2000–2009, based on the remotely-sensed observations from Hugonnet et al. (2021). Dot size represents (a) glacier area and (b) mean elevation. The dashed line indicates the identity line, while the gray line represents the error in geodetic glacier mass balance".*

Figure 8: Highlight in the text that variability between glaciers for simulated mass balance is greater than for geodetic mass balance. What would be a possible explanation?

Agreed. We have highlighted the following in the text:

*"… The scatter plots show significant variability in model performance, with some glaciers (represented by each point) distributed close to the identity line and others deviating significantly. The model simulates a more negative glacier mass balance compared to the geodetic glacier mass balance…"*

Regarding possible explanations, we have added:

*"…The notable variability in model performance (Fig. 7b) could be attributed to inaccuracies in precipitation data for some catchments, as estimating precipitation in high-elevation remote areas remains a complex challenge (Ruelland, 2020)".*

Please note that the numbering of the figures has been adjusted because some were moved to the appendix.

L614-615: Remove or shorten this sentence, as it is likely repeated later. There is no need to refer to the next section in the last sentence of the current section.

Agreed. Sentences were removed.

Fig. 12: add the "mm/year and mm/month" to the numbers in the legend, or alternatively, put the numbers in a separate table would allow to compare them more clearly.

Agreed. We believe that both suggestions help to clarify and better highlight our results. We have added the units to the figure and included a table with the values of the water balance components.

The modified figure is shown here:

[Figure]

Figure 11. Water balance of Lake Titicaca for the period 1982–2016 in terms of (a) interannual variability and (b) seasonal cycle. The values in parentheses correspond to the mean annual or monthly values for the period 1982–2016. The lake water balance follows the equation $P_{lake} + Q_{in} - E_{lake} - Q_{out} - dh/dt + \varepsilon = 0$.

The new table added is as follows:

Table 5. Lake Titicaca water balance components simulated for the period 1994–2003. The lake water balance follows the equation $P_{lake} + Q_{in} - E_{lake} - Q_{out} - dh/dt + \varepsilon = 0$.

| Components | mm yr$^{-1}$ | mm month$^{-1}$ |
|---|---|---|
| $P_{lake}$ | 744 | 62 |
| $Q_{in}$ | 958 | 80 |
| $E_{lake}$ | 1,616 | 135 |
| $Q_{out}$ | 121 | 10 |
| $dh/dt$ | -50 | -4 |
| $\varepsilon$ | -15 | -1 |

Section 4.2.2: It would be useful to include the relative contributions of different water balance terms in the main text, as referred to in the abstract.

Agreed. Previously, we only provided relative water balance values for a few components. We have now included values for both inputs and outputs. Below are the updated details:

*"This means that 44% of the inflows come from direct precipitation over the lake, while the remaining portion (56%) comes from the upstream catchments. Regarding outflows, annual evaporation from the lake is 1,616 mm ($\sigma$ = 28 mm), and the downstream outflow is 121 mm ($\sigma$ = 191 mm). Thus 93% of the losses are due to evaporation and 7% are due to downstream outflow."*

Section 5.1: This section contains significant repetition and should be condensed.

We recognize that this section contained many repetitions and that the new findings need to be clearly differentiated from the study of Lima-Quispe et al. (2021). The first paragraph of this section has been rewritten as follows:

*"5.1 Main findings*

[revised manuscript text omitted]

L736-740: the direct comparison and description of the added value of the present study with Hosseini-Moghari et al., 2020 is a repetition from the intro, and possibly redundant as they treat very different lake system (Urmia), moreover, there are more water balance studies of large lakes, such as Vanderkelen et al., 2020 and other studies included in the introduction.

Agreed. To avoid repetition, we have removed sentences referencing other lake references from different contexts. Instead we have focused on highlighting findings related to Lake Titicaca. The proposed changes are reflected in the previous responses.

L744-750: These lines contrast the following paragraph: here, the point is made the current study includes more detailed representation of snow and ice, and irrigation in the upstream catchments. In the next paragraph there is discussed that these components are of little importance to the water balance of the lake. While the manuscript proves its value of uncovering this, it should not be highlighted this clear in the text that it is an added value.

Agreed. To emphasize the inclusion of these processes, we have formulated a clear research question in the Scope and Objectives section, specifically to understand the sensitivity of water level fluctuations to natural and anthropogenic processes. It is precisely because processes related to snow and ice as well as irrigation were considered that it was possible to show that they had a minimal impact on the variations in the lake's water levels, which are primarily driven by rainfall and evaporation variability. It is worth noting also that, in the face of greater anthropogenic pressures, future irrigation could increase and net irrigation withdrawals may have a greater impact on lake fluctuations. As far as snow and ice processes are concerned, for the Achacachi catchment, the contribution of the total annual inflow (rainfall+snowmelt+ice melt) is 19% for the snowmelt and 7% for the ice melt. This means that snow and ice processes are not negligible in some catchments and at a more local scale even though their impact is limited on lake variations. Finally, we intend to use our integrated model to evaluate irrigation management and climate change scenarios to understand the potential impacts on water level fluctuations. The inclusion of these processes in the development of a water management tool is therefore justified.

Some of the issues were addressed in the Discussion section, highlighting as one of the novelties of the study:

*"Second, through the hydrologic sensitivity analysis, we demonstrate that net irrigation withdrawals and snow and ice melt have minimal impact on lake level fluctuations, indicating that it is primarily driven by rainfall and evaporation variability."*

L878-880: Is it really necessary to refine the glacier simulations as they are not important in the water level simulations?

This not necessary to consider this aspect at present, as its impact on lake fluctuations is minimal. However, it could become more important if we aim to evaluate future changes in glaciers and their potential impacts on specific catchments (e.g. Achacachi and Escoma). To address this, we have modified the sentence to: *"If it is intended to simulate future changes in glaciers, it may be beneficial to include morphometric changes in the model, drawing inspiration from simple approaches in the literature (e.g., Seibert et al., 2018). To initialize the model, global glacier thickness datasets could be used (e.g., Farinotti et al., 2019; Millan et al., 2022)."*

Section 5.3: how feasible is it to translate this modelling chain to another large lake? Also, what are the implications for Lake Titicaca of the presented results? This would be interesting to include in the papers abstract.

Agreed. To address these questions, we have added relevant sentences to Section 5.3 and the Abstract.

For the first question, the following sentence has been included in Section 5.3:

*"…The conceptual models within the modeling framework are easy to apply, require minimal data, and are computationally inexpensive. Several of these models are part of the WEAP platform, which is openly accessible for developing countries (for academic purposes and public institutions). Additionally, we provide in the current article detailed equations for models that are not included in this platform…"*

Regarding the second question, the following sentence was included in Section 5.3:

*"…Contrary to the perceptions of some stakeholders, who often attribute the lake's water level variations to water withdrawals for irrigation or glacier retreat, this study demonstrates that Lake Titicaca's variations are primarily driven by rainfall and evaporation variability…"*

Appendix: Ensure all figures in the appendix are referenced in the main manuscript.

Agreed. We have checked that all the figures in the appendix were referenced in the main text.

**New references**

Allen, R. G., Pereira, L. S., Raes, D., and Smith, M.: Crop evapotranspiration-Guidelines for computing crop water requirements-FAO Irrigation and drainage paper 56, FAO Rome, 300, D05109, 1998.

Hassanzadeh, E., Zarghami, M., and Hassanzadeh, Y.: Determining the Main Factors in Declining the Urmia Lake Level by Using System Dynamics Modeling, Water Resour. Manag., 26, 129–145, https://doi.org/10.1007/s11269-011-9909-8, 2012.

---

## Author Comment (AC2)

**Responses to comments from Dr. Benjamin Kraemer**

On "Assessing the different components of the water balance of Lake Titicaca" by Nilo Lima-Quispe Denis Ruelland, Antoine Rabatel, Waldo Lavado-Casimiro, and Thomas Condom.

**General comments**

Lima-Quispe et al present a valuable contribution to the field of hydrology by addressing the challenging task of assessing the water balance of a poorly-gauged large lake, Lake Titicaca. The integrated modeling framework proposed in the study is novel and addresses limitations in existing methods, particularly for large lakes with sparse data availability. The study's findings on the dominant role of precipitation and evaporation in Lake Titicaca's water balance are substantial and have broader implications for water resource management in similar regions. The scientific methods and assumptions are generally sound, and the results are comprehensive and well-supported.

We would like to sincerely thank Dr. Benjamin Kraemer for the time and effort he dedicated to reviewing the initial manuscript and for providing numerous clear, pertinent, and constructive suggestions for improvement. We are grateful he shared ideas helped to better highlight the novelty and clarity of the paper. For ease of reference, comments are presented in black text, while our responses are in blue, and suggested modifications are highlighted in *blue italics*.

The most important areas of feedback are:

- **Clarity and Focus:** The introduction could benefit from a more research question-driven approach. Clearly stating the main research question at the outset would improve focus and engagement. Additionally, consider streamlining the methods section by moving some of the detailed discussions (e.g., glaciological and hydrological control data) to the appendix.

  Agreed. We addressed these comments carefully. The specific responses and proposed changes can be found in the specific comments under the following criteria: "1. Relevance to HESS Scope", "2. Novelty", and "4. Scientific Methods and Assumptions".

- **Novelty and Contributions:** While the novelty of the integrated modeling framework is evident, it could be further emphasized in the abstract and introduction. Clearly articulating how this approach differs from traditional models and highlighting its unique advantages would strengthen the manuscript's impact.

  Agreed. We addressed these comments carefully. The specific responses and proposed changes can be found in the specific comments under the following criteria: "1. Relevance to HESS Scope", "2. Novelty".

- **Assumptions and Limitations:** The discussion of assumptions, particularly regarding the fixed glacier area and the exclusion of groundwater exchanges, could be expanded. Addressing the potential limitations of these assumptions and their implications for the study's findings would add depth to the analysis.

  Agreed. We addressed these comments carefully. The specific responses and proposed changes can be found in the specific comments under the following criteria: "3. Substantial Conclusions", and "4. Scientific Methods and Assumptions".

- **Incorporating relevant data sources:** Several key data sources were not included but could strengthen the manuscript substantially especially data on irrigated versus non-irrigated agriculture and LSWT.

  Agreed. We addressed these comments carefully. The specific responses and proposed changes can be found in the specific comments under the following criteria: "4. Scientific Methods and Assumptions".

- **Reproducibility:** Providing the modeling code or a detailed description of the algorithms as supplementary material would significantly enhance the reproducibility of the study and its value to the research community.

  Agreed. We addressed these comments carefully. The specific responses and proposed changes can be found in the specific comments under the following criteria: "6. Traceability and Reproducibility".

- **Presentation:** The manuscript is well-written and organized. However, the discussion section could be condensed to avoid repetition and improve clarity. Additionally, minor revisions to figures and captions could enhance their visual appeal and informativeness.

  Agreed. The specific responses and proposed changes can be found in the comments under the following criteria: "10. Overall Presentation".

**Specific comments**

I have organized my feedback below according to the HESS review criteria.

1. Relevance to HESS Scope:

The manuscript addresses topics that are highly relevant within the scope of Hydrology and Earth System Sciences (HESS). It focuses on assessing the water balance of Lake Titicaca using an integrated modeling framework, which is particularly valuable given the challenges of studying poorly gauged large lakes. This research is of significant interest to the hydrology community due to its emphasis on understanding the impacts of climate variability and human activities on lake water levels.

We would like to thank the reviewer for this encouraging comment.

**Suggestions for Improvement:**

- **Lines 123-129:** Consider enhancing the discussion on the relevance of the study by providing specific examples of how the integrated modeling framework addresses gaps in existing research. For example, you could highlight how this study overcomes certain limitations faced by previous research on poorly gauged large lakes. This would help position the manuscript more clearly within the broader context of current literature.

  Agreed. This feedback is very pertinent. To address it, we have reorganized the Introduction by creating a new section titled *"1.1 On the need for an integrated water balance in large lakes"*. In this section, we emphasize that understanding the drivers of large lake fluctuations requires a comprehensive water balance that includes both natural flows and water management. We also clarify the concept of integrated modeling and explain how it differs from traditional approaches.

Additionally, we introduce studies that have attempted to address integrated water balance and their limitations, along with the challenges faced in large lakes.

We have added the following sentences to highlight the need for an integrated water balance that considers both natural flows and water management:

*"…Understanding the main drivers of fluctuations in water levels is crucial for effective lake management, which requires a realistic water balance accounting for both natural processes and anthropogenic pressures (Wurtsbaugh et al., 2017). Several studies on large lakes (>500 km2) (e.g. Rientjes et al., 2011; Vanderkelen et al., 2018; Wale et al., 2009) have estimated water balance under the assumption that net water withdrawals in the contributing catchment are negligible. However, this assumption may no longer be valid due to changing climate conditions and increased competition for water uses, potentially leading to reduced upstream inflow (Wurtsbaugh et al., 2017). For example, Schulz et al. (2020) demonstrated that net withdrawals for irrigation exacerbate the decline in storage at Lake Urmia, which is also impacted by climate change."*

We have also added the following sentences to differentiate integrated modeling from traditional approaches:

*"In large lakes, it is essential to adopt integrated water balance modeling, which represents the interactions and feedbacks between natural hydrologic processes and water management within a single modeling framework (Niswonger et al., 2014). Unlike traditional decision support systems applied to large lakes (Hassanzadeh et al., 2012), which typically simulate natural flows and irrigation water requirements independently, integrated modeling enables these processes to be simulated in a coupled and dynamic manner…"*

2. Novelty:

The manuscript introduces novel concepts, particularly in the integration of various hydrological processes, such as snow and ice melt, heat storage changes in evaporation, and irrigation impacts, within a single modeling framework. However, the manuscript would benefit from a clearer differentiation from previous work.

Agreed. Responses to this comment are addressed below.

**Suggestions for Improvement:**

- **Abstract and Introduction:** The abstract and introduction could be more research question-driven. Clearly identifying the main research question in the first few sentences would improve focus. Consider whether there is uncertainty or disagreement over the dominant drivers of Lake Titicaca's water budget and clarify if this manuscript aims to resolve such questions.

  Agreed.

  In the Abstract, we have identified the knowledge gap in Lake Titicaca as follows:

  *"In the face of climate change and increasing anthropogenic pressures, a reliable water balance is crucial for understanding the drivers of water level fluctuations in large lakes. However, in poorly-gauged hydrosystems such as Lake Titicaca, most components of the water balance are not directly measured. Previous estimates for this lake have relied on scaling factors to close the water balance, which introduces additional uncertainty…"*

In the Introduction, we have added a paragraph that highlights the limitations of previous studies in Lake Titicaca:

*"Unlike other large lakes, very few studies have been conducted on Lake Titicaca. The only study that modeled the water balance of Lakes Titicaca and Poopó was the one by Lima-Quispe et al. (2021) using the Water Evaluation and Planning System (WEAP) platform with a monthly time step for the period 1980–2015. The study aimed to distinguish the relative contributions of climate and irrigation management on water level fluctuations. However, the modeling approach proposed by the authors has a significant limitation because it is based on a scaling factor for precipitation over the lake to close the water balance, which clearly introduces additional uncertainty. Other methodological shortcomings include: (i) the omission of snow and ice processes, which can play a non-negligible role in this high-elevation region; (ii) the estimation of evaporation using the Penman method, without accounting for changes in heat storage; and (iii) the use of historical monthly averages (humidity, wind speed and incoming solar radiation) to calculate reference evapotranspiration and evaporation, without considering interannual variability."*

- **Lines 155-163:** The introduction mentions several research questions. It would be beneficial to identify the most important one and focus on it throughout the introduction, rather than on data sourcing details, which might detract from the main narrative.

  Agreed. To address the comment, we have improved the articulation of our research questions and objectives, focusing on one main question supported by specific questions. The proposed changes are as follows:

  *"In addressing the challenges and limitations of representing hydrologic processes in poorly-gauged large lakes such as Lake Titicaca, we pose the following key question: How can a reliable water balance be estimated? To answer this, we present an integrated modeling framework based on conceptual models to estimate the water balance of Lake Titicaca more reliably. The modeling framework is applied at a daily time step for the period 1982–2016, allowing us to represent water level fluctuations over a wide range of hydroclimatic conditions. The specific questions are: To what extent are water level variations sensitive to net irrigation withdrawals and to snow and ice processes? What is the role of heat storage change in evaporation from the lake? To address these questions, new approaches are introduced for: (i) predicting upstream inflow, including hydrologic sensitivity to net irrigation consumption and snow and ice processes; and (ii) estimating evaporation from the lake using the Penman method, while accounting for changes in heat storage."*

- **Lines 161-163:** Please clarify what is meant by an "integrated" water balance approach and how it differs from traditional models. This will help readers better appreciate the novelty of your approach.

  Agreed. To address this comment, we have added a paragraph defining the integrated water balance. *"In large lakes, it is essential to adopt integrated water balance modeling, which represents the interactions and feedbacks between natural hydrological processes and water management within a single modeling framework (Niswonger et al., 2014). Unlike traditional decision support systems applied to large lakes (Hassanzadeh et al., 2012), which typically simulate natural flows and irrigation water requirements independently, integrated modeling enables these processes to be simulated in a coupled and dynamic manner"*

3. Substantial Conclusions:

The conclusions presented in the manuscript are well-supported by the results. The inclusion of snow and ice processes and heat storage changes in evaporation is particularly important for modeling lake water levels.

*Thank you very much. Responses to this comment are addressed below.*

**Suggestions for Improvement:**

- **Lines 744-750:** While it is important to highlight that snow, ice, and irrigation have minimal impacts, ensure this does not detract from the value of including these processes in the model. Emphasizing that understanding the minimal impact is itself a valuable finding could strengthen this point.

  *Agreed. To emphasize the inclusion of these processes, we have formulated a clear research question in the Scope and Objectives section, specifically to understand the sensitivity of water level fluctuations to natural and anthropogenic processes. We agree with the reviewer that it is precisely because processes related to snow and ice as well as irrigation were considered that it was possible to show that they had a minimal impact on the variations in the lake's water levels, which are primarily driven by rainfall and evaporation variability. It is also worth noting that, in the face of greater anthropogenic pressures, future irrigation could increase and net irrigation withdrawals may have a greater impact on lake fluctuations. As far as snow and ice processes are concerned, for the Achacachi catchment, the contribution of the total annual inflow (rainfall+snowmelt+ice melt) is 19% for the snowmelt and 7% for the ice melt. This means snow and ice processes are not negligible in some catchments and at a more local scale even though their impact is limited on lake variations. Finally, in a future study, we intend to use our integrated model to evaluate irrigation management and climate change scenarios to understand their potential impacts on water level fluctuations. The inclusion of these processes in the development of a water management tool is therefore justified.*

  *Some of the issues were addressed in the Discussion section, highlighting as one of the novelties of the study:*

  *"Second, through the hydrologic sensitivity analysis, we demonstrate that net irrigation withdrawals and snow and ice melt have minimal impact on lake level fluctuations, indicating that it is primarily driven by rainfall and evaporation variability."*

- **Lines 377-378:** Elaborate on the potential impact of assuming a fixed glacier area on long-term simulations. Discuss how this assumption might influence the results under different climate scenarios or with ongoing glacier retreat.

  *Agreed. As we are working with the period going from 1982 to 2016, the year of the used glacier inventory (i.e. 2000) falls almost in the mid-range. Over this study period, the general trend of tropical glacier is shrinking (e.g. Masiokas et al., 2020; Rabatel et al., 2013; Vuille et al., 2018). Therefore, the use of a fixed glacier area leads to an underestimation of glacier melt in the period 1982–1999 and an overestimation in the period 2001–2016. The ice stock variation shows a negative value (-5 mm yr-1), indicating a glacier retreat in the period 1982–2016, which is consistent with the effects of global warming and in agreement with in-situ observations (e.g. Rabatel et al., 2013). On average, the ice stock variation was -4.4 mm yr-1 in the period 1982–1999 and -5.5 mm yr-1 in the period 2001–2016, indicating a slightly accelerated melting in the last decades. In addition, most of the glaciers in the study area are small and located at relatively low*

elevations (below 5,500 m a.s.l. in most of the cases). They have already passed the "peak water" (maximum meltwater contribution) and many are about to disappear. At the scale of the whole Titicaca catchment a recent study indicates a peak water situated between 2023 and 2028 depending of the glaciological model used (Wimberly et al., 2024). In such conditions glacier melt tends to be overestimated when assuming a fixed area. By using the glacier area of an intermediate year, the under- or overestimation associated with assuming a fixed area is modulated to some extent. For example, if we had used the 1982 glacier area, the overestimation of melt would have been greater in the later years of the period. Conversely, if we were to use the 2016 area, there would be a significant underestimation in the early years of the modeling period.

Regarding the hydrological importance of the glacier, at the scale of Lake Titicaca, the long-term (1982–2016) annual average contribution of ice melt is insignificant, representing only 1% of the total inflow (rainfall + snowmelt + ice melt). However, in some catchments (Escoma and Achacachi) that show the most important glacial coverage (3.4 and 6.6%, respectively), this percentage is relatively important, reaching about 7% of the total annual inflow. Although our simulations show limited impact of ice melt at the scale of Titicaca lake, it may have more impact at the sub-catchment level.

To evaluate long-term climate and water management scenarios, it is feasible to dynamically simulate both glacier area and volume evolution, since glacier area and ice thickness data are available for the beginning of the 21st century (Farinotti et al., 2019; Millan et al., 2022) to initialize the model.

To address the reviewer's comment, we propose to add the underlined sentences:

*"On the other hand, the estimated glacier mass balances did not consider changes in glacier area, thickness and volume over time. The variation in ice stock over the period 1982–2016 is negative (-5 mm yr-1), which is consistent with the effects of global warming and in agreement with in-situ observations (Rabatel et al., 2013). As the surface area of the glaciers in our model was obtained in 2000 and considering glacier shrinkage worldwide, melting before the year 2000 may be underestimated, and after the year 2000, it may be overestimated. However, the biases are limited to some extent by the choice of an intermediate glacier area for the modeled period. If it is intended to simulate future changes in glaciers, it may be beneficial to include morphometric glacier changes in the model, drawing inspiration from simple approaches in the literature (e.g., Seibert et al., 2018). To initialize the model, global glacier thickness datasets could be used (e.g., Farinotti et al., 2019; Millan et al., 2022)."*

4. Scientific Methods and Assumptions:

The scientific methods and assumptions are clearly outlined and valid. However, the methods section could be made more concise, as it currently includes a level of detail that may be repetitive.

Thank you very much. Responses to the feedback are addressed below.

**Suggestions for Improvement:**

- **Lines 258-259:** The difference between rainfed and irrigated agriculture can also be captured by other land cover datasets, such as those produced by the ESA Land Cover CCI, which offers annual datasets.

Thank you for the suggestion. During the implementation phase of our modeling approach, we reviewed the ESA Land Cover CCI dataset, which provides data from 1992 to present. Unfortunately, this dataset does not identify irrigated land in our study area. As shown in the images below, according to the ESA LC CCI legend, irrigated areas are coded with 20, but this category is absent from the map.

ESA Land Cover CCI for 2016:

[Figure]

Legend (https://datastore.copernicus-climate.eu/documents/satellite-land-cover/WP2-FDDP-LC-2021-2022-SENTINEL3-300m-v2.1.1_PUGS_v1.1_final.pdf):

*Table 1-2: Level 1 (or global) legend of the LC maps, based on the UN-LCCS.*

| Value | Label | Color | RGB |
|-------|-------|-------|-----|
| 0 | No Data | | 0, 0, 0 |
| 10 | Cropland, rainfed | | 255, 255, 100 |
| 20 | Cropland, irrigated or post-flooding | | 170, 240, 240 |
| 30 | Mosaic cropland (>50%) / natural vegetation (tree, shrub, herbaceous cover) (<50%) | | 220, 240, 100 |
| 40 | Mosaic natural vegetation (tree, shrub, herbaceous cover) (>50%) / cropland (<50%) | | 200, 200, 100 |
| 50 | Tree cover, broadleaved, evergreen, closed to open (>15%) | | 0, 100, 0 |
| 60 | Tree cover, broadleaved, deciduous, closed to open (>15%) | | 0, 160, 0 |
| 70 | Tree cover, needleleaved, evergreen, closed to open (>15%) | | 0, 60, 0 |
| 80 | Tree cover, needleleaved, deciduous, closed to open (>15%) | | 40, 80, 0 |

- **Lines 445-446:** Consider using Lake Surface Water Temperature (LSWT) data available from remote sensing sources, such as the ESA Lakes Climate Change Initiative, which might provide more accurate results in a hydrosystem driven by net radiation and evaporation.

In this study we did not directly use remotely sensed LSWT data because our analysis period (1982–2016) extends beyond the coverage of available datasets. For example, ARC-Lake V3 covers 1995 to 2012, and the ESA Lake CCI spans 1995 to 2023. To obtain a daily time series for the entire period, we simulated the LSWT using the Air2Water conceptual model (with air temperature as input). We then calibrated and evaluated the model against ARC-Lake V3 data, achieving acceptable results, as shown in the following figure.

[Figure]

*Figure B1. Lake surface water temperature (LSWT) simulated with the Air2wateR model (Piccolroaz et al., 2013; Toffolon et al., 2014) for Lake Titicaca and its calibration and evaluation performances.*

To assess the reliability of ARC-Lake V3, we compared it with on-site measurements from the OLT (https://olt.geovisorumsa.com/). As shown in the following figure, both datasets exhibit fluctuations within the same range.

[Figure]

*Figure B2. Lake surface water temperature (LSWT) obtained from the buoy site (16.25°S, 68.68°W) in Arc-Lake V3 (MacCallum and Merchant, 2012) and OPLT (Lazzaro et al., 2021).*

In addition, both ARC-Lake and ESA Lake CCI use similar optimal estimation techniques for LSWT retrieval (https://confluence.ecmwf.int/pages/viewpage.action?pageId=348800012).

To address this comment, we have added the following to the Methods section: *"Since there are no long-term measurements of lake surface water temperature (LSWT) and the remotely sensed data sets do not cover the entire study period, the Air2Water model (Piccolroaz et al., 2013; Toffolon et al., 2014) was used to simulate LSWT. Calibration and evaluation were performed against ARC-Lake V3 remotely sensed data (MacCallum and Merchant, 2012) (see Appendix B)."*

- **Lines 295-315:** Consider moving the detailed discussion of glaciological and hydrological control data to the appendix. This would streamline the methods section and enhance readability.

  Agreed. In response to this suggestion, we have moved several sections to the Appendix: the paragraph on glaciological monitoring data specific to the Zongo Glacier; the paragraph in the Methods section on obtaining snow and ice model parameters from the Zongo catchment; and the evaluation of ERA5-Land data, along with details on the LSWT simulation procedure.

- **Lines 469-474:** Discuss the implications of excluding groundwater exchanges, especially in light of observed discrepancies in the water balance. Including a brief sensitivity analysis or discussion on the potential significance of this exclusion under different hydrological conditions could add depth to the analysis.

  Agreed. We acknowledge that we did not discuss the implications of neglecting net groundwater flow in the originally submitted version. According to our results, the long-term water balance has

a minimal closure error (-15 mm yr⁻¹). A possible hypothesis is that this error corresponds to the net groundwater flow, although it is important to note that it also includes the uncertainty associated with the other components of the water balance. The error term indicates a seasonal variation with a positive pattern during the rainy season and a negative pattern during the dry season, suggesting that the lake-groundwater interaction varies seasonally (see following Figure, Figure 11b of the manuscript).

[Figure]

*Figure 11. Water balance of Lake Titicaca for the period 1982–2016 in terms of (a) interannual variability and (b) seasonal cycle. The values in parentheses correspond to the mean annual or monthly values for the period 1982–2016. The lake water balance follows the equation $P_{lake} + Q_{in} - E_{lake} - Q_{out} - dh/dt + \varepsilon = 0$.*

The water level of Lake Titicaca varies by an average of 67 cm throughout the hydrological year, reaching a maximum in April and a minimum in December. The error term shows the highest positive values between December and January, which could indicate a discharge of groundwater into the lake. When the lake reaches high water levels, losses to groundwater tend to dominate. These dynamics suggest that there could be a reversal of the hydraulic gradient during the year, depending on the water level of the lake and the groundwater. Thus, the long-term seasonal water balance is affected and the seasonal simulations are not perfect (see the following Figure).

[Figure]

To address this in the manuscript, we suggest adding the following paragraph in the Discussion section:

*"Figure 11b shows that the error term exhibits a seasonal variation, being positive during the rainy season and predominantly negative during the dry season. Linking the error term to net groundwater flow suggests that groundwater-lake interactions are seasonally variable. Lake water levels fluctuate by an average of 67 cm over the hydrological year, reaching a maximum in April and a minimum in December. In Figure 11b, the error term has the highest positive values between December (26 mm) and January (54 mm), indicating a gain in net groundwater flow to the lake. When the lake reaches high water levels, losses to groundwater tend to dominate. This dynamic suggests that there could be a reversal of the hydraulic gradient throughout the year depending on the water level of the lake and the groundwater. However, it is important to note that the error term reflects not only the net groundwater flow, but also the uncertainty in estimating the other components of the water balance."*

5. Sufficiency of Results:

The results are sufficient to support the interpretations and conclusions. The comprehensive simulations and sensitivity analyses presented strengthen the validity of the findings.

Thank you very much.

6. Traceability and Reproducibility:

The manuscript provides detailed descriptions of datasets, model parameters, and calibration processes, which enhances the traceability of the results. However, reproducibility could be further improved.

Thank you very much. Responses to this comment are addressed below.

**Suggestions for Improvement:**

- **Lines 320-330:** Consider providing the modeling code or at least a detailed description of the algorithms used as supplementary material. Alternatively, making the code available on a public repository would greatly enhance the reproducibility of the study and its usefulness to other researchers.

  Agreed. Most of the models used are part of the WEAP platform; a detailed description of the equations can be found at this link (https://www.weap21.org/webhelp/index.html) and in the article by Yates et al. (2005). The models used for lake evaporation and snow and ice processes are not integrated into WEAP, so detailed equations are provided in the Methods section. These models were implemented in WEAP using the user-defined variables option. WEAP is openly accessible in developing countries for academic purposes and for public institutions.

  To address this comment, we propose to add the following section: *"Code availability. SMM (https://www.weap21.org/webhelp/Two-bucket_Method.htm) and the lake water balance model (https://www.weap21.org/WebHelp/River_Reservoir_Flows.htm) are part of the WEAP platform (https://www.weap21.org/). The models used for snow processes and lake evaporation are not part of WEAP; therefore, the detailed equations are presented in the Methods section. These models were implemented in WEAP using the user-defined variables (https://www.weap21.org/webhelp/User_Defined_Variables.htm)."*

7. Credit to Related Work:

The manuscript appropriately credits related work and clearly indicates the authors' contributions. The review of relevant literature is comprehensive, situating the study within the broader context of hydrological research. However, in some areas, differentiation from previous work could be more explicit.

Thank you very much. To address the feedback, we have clearly differentiated our modeling approach from that of the study by Lima-Quispe et al. (2021). This distinction is specifically addressed in the Introduction, Methods, and Discussion sections (see above for answers and modifications made to the manuscript).

8. Title Accuracy:

The current title, "Assessing the Different Components of the Water Balance of Lake Titicaca," accurately reflects the contents of the paper. However, a more specific title that highlights the key findings could be more informative. Consider options like:

- "Modeling Lake Titicaca's Water Balance: The Dominant Roles of Precipitation and Evaporation."

- "Precipitation and Evaporation as Primary Drivers of Lake Titicaca's Water Balance."

Thank you very much for the suggestion. We agree that the suggested titles better highlight our key findings. We have chosen the first option.

9. Abstract Quality:

The abstract effectively summarizes the key objectives, methods, and findings of the study. However, it could be slightly improved by emphasizing the study's contributions to existing knowledge and the novelty of the integrated modeling framework.

Thank you very much. Responses to the feedback are provided below.

**Suggestions for Improvement:**

- **First 3 Lines:** A clear statement of the research question or gap in the first three lines would significantly improve the abstract's clarity and focus.

  Agreed. We reviewed the abstract and incorporated the suggested modifications. The following sentences were added at the beginning of the abstract: *"In the face of climate change and increasing anthropogenic pressures, a reliable water balance is crucial for understanding the drivers of water level fluctuations in large lakes. However, in poorly-gauged hydrosystems such as Lake Titicaca, most components of the water balance are not directly measured. Previous estimates for this lake have relied on scaling factors to close the water balance, which introduces additional uncertainty….."*

10. Overall Presentation:

The manuscript is well-structured and clear, with a logical flow of information. The figures and tables are well-designed and contribute effectively to the presentation of results.

Thank you very much. Responses to the feedback are provided below.

**Suggestions for Improvement:**

- **Section 5.1 (Discussion):** This section contains significant repetition. It could be condensed by focusing on summarizing key findings without reiterating points made earlier.

Agreed. We have substantially reduced the discussion in Section 5.1 and eliminated the redundant sentences previously mentioned. In the first paragraph of Section 5.1, we have highlighted and summarized the new findings of this study. We propose the following modification:

*"5.1 Main findings*

[revised manuscript text omitted]

- **Lines 568-573:** These sentences could be made more concise by merging them into a single, impactful sentence.

  Agreed. We have refined the writing to make it more concise. The modified sentences are as follows: *"...The scatter plots reveal significant variability in model performance, with some glaciers (represented by each point) close to the identity line and others deviating significantly. The model simulates a more negative glacier mass balance compared to the geodetic glacier mass balance. Figure 7a displays glaciers according to their surface area, while Figure 7b shows them based on mean elevation. ..."*.
  Please note that the numbering of the figures has been updated due to some being moved to the appendix.

11. Language Fluency and Precision:

The language used in the manuscript is fluent and precise, though it could be made more concise. The manuscript is generally well-written, with minimal grammatical errors or ambiguities, and the technical terminology is used appropriately.

Thank you very much. Your specific comments are addressed below.

**Suggestions for Improvement:**

- **Lines 124-126:** Clarify whether "net water withdrawals" refer specifically to the lake or the entire hydrological system. This could be rephrased for clarity: "net water withdrawals from both the lake and its contributing catchments."

  Agreed. In this case, it refers to the contributing catchments. The sentence has been modified to: *"...Several studies on large lakes (>500 km²) (e.g. Rientjes et al., 2011; Vanderkelen et al., 2018; Wale et al., 2009) have estimated the water balance under the assumption that net water withdrawals in the contributing catchments are negligible..."*

- **Lines 614-615:** Consider removing or shortening this sentence, as it may be repetitive. There is no need to refer to the next section in the last sentence of the current section.

  Agreed. Sentences were removed.

- **Lines 745-747:** To avoid redundancy, consider rephrasing to: "this study enhances the representation of snow and ice processes and irrigation impacts."

  Agreed. To avoid redundancy, the paragraph containing this sentence has been rewritten. Now the idea is written as follows: *"...Our modeling approach benefits from: (i) a rigorous calibration and evaluation procedure for simulating upstream inflow, including hydrologic sensitivity to net irrigation consumption and to snow and ice processes;..."*

- **Lines 807-808:** Consider omitting this sentence as it might be unnecessary.

  Agreed. Sentences were removed. Now the "Limitations of the modeling framework" section starts with the following sentence: *"Model forcing and evaluation data are the main sources of uncertainty…"*

12. Mathematical Formulae and Symbols:

Mathematical formulae, symbols, abbreviations, and units appear to be correctly defined and used throughout the manuscript. The equations are clearly presented and appropriately referenced within the text.

Thank you very much.

13. Clarity of Figures and Tables:

The figures and tables are clear and informative. They effectively support the text and enhance the reader's understanding of the study's findings.

Thank you very much, comments are addressed below.

**Suggestions for Improvement:**

- **Figure 2, Figure 8:** Reverse the order of gradient legends so that low values are on the bottom and high values on the top.

  Agreed. The order of the legend values was reversed. Here are the modified figures:

  Figure 2

[Figure]

*Figure 2. Spatial distribution of (a) annual precipitation and (b) mean temperature for the hydrological period 1980–2016 according to GMET (Gridded Meteorological Ensemble Tool).*

Figure 8

[Figure]

*Figure 7. Scatter plots comparing simulated and geodetic glacier mass balance for 2000–2009, based on the remotely-sensed observations from Hugonnet et al. (2021). Dot size represents (a) glacier area and (b) mean elevation. The dashed line indicates the identity line, while the gray line represents the error in geodetic glacier mass balance.*

- **Figure 2, Right Panel:** Adjust the color bar to ensure that white corresponds to zero, or use a sequential color bar to improve the visual interpretation of the data.

  Agreed. To address the reviewer's comment, the figure has been modified using diverging colors, with yellow corresponding to zero Celsius. See the modifications above.